# Asynchronous Speedup in Decentralized Optimization

**Mathieu Even**[(1)] **, Hadrien Hendrikx**[(2)] **, Laurent Massoulié**[(1,3)] **,**
[(1)]Inria-Paris, DI ENS and PSL Research University
[(2)]Ecole Polytechnique Fédérale de Lausanne (EPFL)
[(3)]MSR-Inria Joint Center

## Abstract

In decentralized optimization, nodes of a communication network each possess a local objective function, and communicate using gossip-based methods in order to minimize the average of these per-node functions. While synchronous algorithms are heavily impacted by a few slow nodes or edges in the graph (the *straggler problem*), their asynchronous counterparts are notoriously harder to parametrize. Indeed, their convergence properties for networks with heterogeneous communication and computation delays have defied analysis so far. In this paper, we use a *continuized* framework to analyze asynchronous algorithms in networks with delays. Our approach yields a precise characterization of convergence time and of its dependency on heterogeneous delays in the network. Our continuized framework benefits from the best of both continuous and discrete worlds: the algorithms it applies to are based on event-driven updates. They are thus essentially discrete and hence readily implementable. Yet their analysis is essentially in continuous time, relying in part on the theory of delayed ODEs. Our algorithms moreover achieve an *asynchronous speedup*: their rate of convergence is controlled by the eigengap of the network graph weighted by local delays, instead of the network-wide worst-case delay as in previous analyses. Our methods thus enjoy improved robustness to stragglers.

## 1 Introduction

We study the following optimization problem:

$$\min_{x \in \mathbb{R}^d} \left\{ f(x) = \sum_{i=1}^n f_i(x) \right\}, \tag{1}$$

where each individual function $f_i : \mathbb{R}^d \to \mathbb{R}$ for $i \in [n]$ is held by an agent $i$. We consider *asynchronous* and *decentralized* optimization methods that do not rely on a central coordinator. This is particularly relevant in large-scale systems in which centralized approaches suffer from a communication bottleneck at the central controller. Decentralized optimization is relevant to supervised learning of models in data centers, but also to more recent federated learning scenarios where data and computations are distributed among agents that do not wish to share their local data. We focus on asynchronous operations because of their scalability in the number of agents in the system, and their robustness to node failures and to *stragglers*. In the case of empirical risk minimization, $f_i$ represents the empirical risk for the local dataset of node $i$, and $f$ the empirical risk over all datasets. Another important example, that plays the role of a toy problem for both decentralized and/or stochastic optimization is that of network averaging, corresponding to $f_i(x) = \|x - c_i\|^2$ where $c_i$ is a vector attached to node $i$. In this case, the solution of Problem (1) reads $\bar{c} = \frac{1}{n} \sum_{i=1}^n c_i$.

We assume that agents are located at the nodes of a connected, undirected graph $G = (V, E)$ with node set $V = [n]$. An agent $i \in V$ can compute first-order quantities (gradients) related to its

Workshop on Federated Learning: Recent Advances and New Challenges, in Conjunction with NeurIPS 2022 (FL-NeurIPS'22). This workshop does not have official proceedings and this paper is non-archival.

local objective function $f_i$, and can communicate with any adjacent agent in the graph. Our model of asynchrony derives from the popular randomized gossip model of [5]. In this model, nodes update their local values at random activation times using pairwise communication updates. This asynchronous model makes the idealized assumption of instantaneous communications, and hence does not faithfully represent practical implementations. To alleviate this drawback, several works [2, 23, 24, 34, 37, 39] introduce communication and computation delays in either pairwise updates, or in asymmetric gossip communications. However, all these works provide convergence guarantees that either require global synchronization between the nodes, or are implicitly determined by an upper bound on the worst-case delay in the whole graph. Indeed they assume that i) for some $k_{\max} > 0$, for all edges $(ij) \in E$, each communication between agents $i$ and $j$ overlaps with at most $k_{\max}$ other communications in the whole graph, and ii) either agents $i$ or graph edges $(ij)$ are activated for agent interaction sequentially in an i.i.d. manner. Thus assuming distributed asynchronous operation where individual nodes schedule their interactions based only on local information, the $k_{\max}$ constraint can only be enforced by requiring individual nodes to limit their update frequency to $1/(n\tau_{\max})$.

Consequently, the resulting algorithms have temporal convergence guarantees proportional to $\tau_{\max}$. They are thus not robust to *stragglers*, i.e. slow nodes or edges in the graph that induce large $\tau_{\max}$. To understand the scope for improvement over such methods, recall that for synchronous algorithms with updates performed every $\tau_{\max}$ seconds, for $L$-smooth and $\sigma$-strongly convex functions $f_i$, the time required to reach precision $\varepsilon > 0$ for $\frac{1}{n}\sum f_i$ is lower-bounded by [32]:

$$\Omega\left(\tau_{\max}\mathrm{Diam}(G)\sqrt{\kappa}\ln(\varepsilon^{-1})\right), \tag{2}$$

where $\kappa = L/\sigma$ is the condition number of the functions $f_i$ and $\mathrm{Diam}(G)$ is the diameter of graph $G$. In this article we seek better dependency on individual delays in the network. Specifically we consider the following

**Assumption 1** (Heterogeneous delays). *There exist $\tau_{ij}$ for $(ij) \in E$ and $\tau_i^{\mathrm{comp}}$ for $i \in V$ such that communications between two neighboring agents $i$ and $j$ in the graph take time at most $\tau_{ij}$, and a computation at node $i$ takes time at most $\tau_i^{\mathrm{comp}}$.*

Under such heterogeneous delay assumptions, **how robust to stragglers can decentralized algorithms be?** One can adapt the proof of [32] to Assumption 1 to generalize (2) as:

$$\Omega\left(D(\tau)\sqrt{\kappa}\ln(\varepsilon^{-1})\right), \tag{3}$$

where $D(\tau) = \sup_{(i,j)\in V^2} \mathrm{dist}(i,j)$ for $\mathrm{dist}(i,j) = \inf\left\{\tau_i^{\mathrm{comp}} + \tau_j^{\mathrm{comp}} + \sum_{k=0}^{p-1} \tau_{i_k i_{k+1}}\right\}$, where the inf is taken over all $(i = i_0, \ldots, i_p = j) \in V^{p+1}$ that verify $\forall 1 \leqslant k \leqslant p$, $(i_k, i_{k+1}) \in E$.

Here $\mathrm{dist}(i,j)$ is the time distance between nodes $i$ and $j$, and $D(\tau)$ is the *diameter* of graph $G$ for this distance. $D(\tau)$ is the generalization of $\tau_{\max}\mathrm{Diam}(G)$ to the heterogeneous-delay setting. This lower bound suggests that robustness to stragglers is possible: indeed if a fraction of the nodes or edges is too slow (large delay $\tau_{ij}$), this may not even impact this lower bound, since the shortest path between two nodes may always take another route. We aim at building *decentralized* algorithms with performance guarantees that enjoy such robustness to individual delay bounds. However, since we focus on fully decentralized algorithms, our performance guarantees will not be expressed in terms of some diameter $D(\tau)$ as in (3) but instead in terms of some spectral characteristics of the graph at hand[1]. We thus seek performance guarantees similar to (3) with in place of $D(\tau)$ the term $\lambda_2(\Delta_G(\nu))^{-1}$ for some parameters $\nu_{ij}$ that depend on delay characteristics local to edge $(ij)$, where the graph Laplacian $\Delta_G$ is defined as follows.

**Definition 1** (Graph Laplacian). *Let $\nu = (\nu_{ij})_{(ij)\in E}$ be a set of non-negative real numbers. The Laplacian of the graph $G$ weighted by the $\nu_{ij}$'s is the matrix $\Delta_G(\nu)$ with $(i,j)$ entry equal to $-\nu_{ij}$ if $(ij) \in E$, $\sum_{k\sim i} \nu_{ik}$ if $j = i$, and $0$ otherwise. In the sequel $\nu_{ij}$ always refers to the weights of the Laplacian, and $\lambda_2(\Delta_G(\nu))$ denotes this Laplacian's second smallest eigenvalue.*

---

[1] Note that similar spectral characteristics (albeit based on a single worst-case delay parameter $\tau_{\max}$) appear in [2, 23, 32, 34, 37, 39].

## 2 Contributions and related works

### 2.1 Contributions

We consider the *network averaging problem*, for which we introduce *Delayed Randomized Gossip* in Section 3. Building on recent works on continuized gradient descent for Nesterov acceleration [12], we analyze Delayed Randomized Gossip in the continuized framework, that allows a continuous-time analysis of an algorithm even though the latter is based on discrete, hence practically implementable operations. Our analysis leads to explicit stability conditions that have the appealing property of being *local*, i.e. they require each agent to tune its algorithm parameters to delay bounds in its graph neighborhood. These conditions ensure a linear rate of convergence determined by $\lambda_2\big(\Delta_G(\nu)\big)$, for weights of order $\nu = 1/(\sum_{(kl)\sim(ij)} \tau_{kl})^2$. This dependency of weights in the Laplacian on local delay bounds is what we call the *asynchronous speedup*, since it implies a scaling that is no longer proportional to $\tau_{\max}$.

We provide extensions in Section 4 of the delayed randomized gossip we introduce next, to the more general decentralized optimization problem defined in Equation (1), and generalize our setup to capacity constraints, beyond the *Poisson point process* assumption, in Section 5, that both further illustrate the generality of our approach. We also highlight a phenomenon reminiscent of Braess' paradox in Section 6: sparsifying the graph may accelerate convergence speed.

### 2.2 Related works

*Decentralized Optimization and Gossip Algorithms.* Gossip algorithms [5, 9] were initially introduced to compute the global average of local vectors with local pairwise communications only (no central coordinator), and were generalized to decentralized optimization. Two types of gossip algorithms appear in the literature: synchronous ones, where all nodes communicate with each other simultaneously [9, 19, 32, 33], and randomized ones [5, 28]. A third category considers directed (non-symmetric) communication graphs [2, 40] which are much easier to implement asynchronously. In the synchronous framework, the communication speed is limited by the slowest node (*straggler* problem), whereas the classical randomized gossip framework of [5] assumes communications to happen instantaneously, and thus does not address the question of how to deal with delays. [2, 23, 34, 37, 39] introduce delays in the analysis of decentralized algorithms; as mentioned in the introduction, their analyses and algorithms are not robust to stragglers, relying on a single upper bound on the delays of all edges. [38] study how sparsifying the communication graph can lead to faster decentralized algorithm. Their approach is different from ours in Section 6: they do not consider asynchronous algorithms with physical constraints (delays and capacity), but synchronous algorithms where sequentially matchings are built in the graph. Yet, we observe similar phenomenon as theirs in Section 6. We refer the reader to [29] for a more complete survey of gossip algorithms.

*Handling Asynchrony.* The dynamics of asynchronous optimization algorithms are significantly more complex than their synchronous counterparts. Their study goes back to the monograph of [4], where asynchrony is modelled through a global ordering of events, providing the formalism classically used. Most of the recent literature is then derived from a distributed asynchronous variant of SGD called *HOGWILD!* [31]. [22, 25] introduce alternative orderings of the iterates (before-read and after-read) that correspond to different views of the same sequence of updates, and which simplify the analysis through the use of perturbed or virtual iterates [14, 25, 27, 35, 42], even though proofs and convergence guarantees under realistic assumptions on the intricacies between iterates, delays and choices of coordinates, are a challenging problem [7, 36]. We refer the interested reader to [1] for a more exhaustive survey of advances in asynchrony, both in shared-memory and decentralized models. In this paper, we deal with asynchrony and delays from a different viewpoint: the analysis is inspired by time-delayed ODE systems [30], and the assumptions related to delays and asynchrony (such as Assumption 1) do not need to be translated into discrete-time ones, as in the above references. Finally, we believe our continuous time framework to be particularly adequate for the study and design of asynchronous algorithms, in the decentralized setting as in this paper, but also in centralized settings where it may remove the need to introduce a discrete ordering of events and thus avoid difficulties that lead to unrealistic assumptions, such as the *after/before-read* approaches [22].

---

[2] We write $(ij) \sim (kl)$ and say that two edges $(ij)$, $(kl)$ are neighbors if they share at least one node.

# 3 Delayed Randomized Gossip for Network Averaging

## 3.1 Randomized gossip

Let $G = (V, E)$ be a connected graph on the set of nodes $V = [n]$, representing a communication network of agents. Each agent $i \in V$ is assigned a real vector $x_0(i) \in \mathbb{R}^d$. The goal of the averaging (or gossip) problem is to design an iterative procedure allowing each agent in the network to estimate the average $\bar{x} = \frac{1}{n} \sum_{i=1}^n x_0(i)$ using only local communications, *i.e.*, communications between adjacent agents in the network.

In randomized gossip [5], time $t$ is indexed continuously by $\mathbb{R}^+$. A Poisson point process [18] (abbreviated as *P.p.p.* in the sequel) $\mathcal{P} = \{T_k\}_{k \geqslant 1}$ of intensity $I > 0$ on $\mathbb{R}^+$ is generated: $T_0 = 0$ and $(T_{k+1} - T_k)_{k \geqslant 0}$ are *i.i.d.* exponential random variables of mean $1/I$. For positive intensities $(p_{ij})_{(ij) \in E}$ such that $\sum_{(ij) \in E} p_{ij} = I$, for every $k \geqslant 0$, at $T_k$ an edge $(i_k j_k)$ is *activated* with probability $p_{i_k j_k}/I$, upon which adjacent nodes $i_k$ and $j_k$ communicate and perform a pairwise update. The *P.p.p.* assumption implies that edges are activated independently of one another and from the past: the activation times of edge $(ij)$ form a *P.p.p.* of intensity $p_{ij}$.

To solve the gossip problem, [5] proposed the following strategy: each agent $i \in V$ keeps a local estimate $x_t(i)$ of the average and, upon activation of edge $(i_k j_k)$ at time $T_k \in \mathbb{R}^+$, the activated nodes $i_k, j_k$ average their current estimates:

$$x_{T_k}(i_k),\ x_{T_k}(j_k) \longleftarrow \frac{x_{T_k-}(i_k) + x_{T_k-}(j_k)}{2} . \tag{4}$$

Writing $f(x) = \sum_{(ij) \in E} \frac{p_{ij}}{I} f_{ij}(x)$, for $f_{ij}(x) = \frac{1}{2} \|x(i) - x(j)\|^2$ and $x = (x(i))_{i \in V}$, [12] observe that local averages (4) correspond to stochastic gradient steps on $f$:

$$x_{T_k} \longleftarrow x_{T_k-} - \frac{K_{i_k j_k}}{p_{i_k j_k}} \nabla f_{i_k j_k}(x_{T_k-}), \tag{5}$$

for step sizes $K_{i_k j_k} = \frac{p_{i_k j_k}}{2}$. Yet, this continuous-time model with *P.p.p.* activations implicitly assumes instantaneous communications, or some form of waiting. Indeed, the gradient is computed on the current value of the parameter, which is $x_{T_k-}$. In the presence of (heterogeneous) communication delays (Assumption 1), a more realistic update uses the parameter $x_{S_k}$ at a previous time $S_k < T_k$, to account for the time it takes to compute and communicate the gradient. In this case, the updates write

$$x_{T_k} \longleftarrow x_{T_k-} - \frac{K_{i_k j_k}}{p_{i_k j_k}} \nabla f_{i_k j_k}(x_{S_k}). \tag{6}$$

## 3.2 The continuized framework

Our approach uses the **continuized framework** [12], which amounts to consider continuous-time evolution of key quantities, with discrete jumps at the instants of Poisson point processes. This gives the best of both continuous (for the analysis and assumptions) and discrete (for the implementation) worlds. From now on and for the rest of the paper, we assume that Assumption 1 holds.

Edges $(ij) \in E$ locally generate independent *P.p.p.* $\mathcal{P}_{ij}$ of intensity $p_{ij} > 0$ (random activation times, with *i.i.d.* intervals, exponentially distributed with mean $1/p_{ij}$). As mentioned previously, $\mathcal{P} = \bigcup_{(ij) \in E} \mathcal{P}_{ij}$ is a *P.p.p.* of intensity $I = \sum_{(ij) \in E} p_{ij}$, and noting $\mathcal{P} = \{T_1 < T_2 < \dots\}$, at each clock ticking $T_k$, $k \geqslant 1$, an edge $(i_k j_k)$ is chosen with probability $p_{i_k j_k}/I$. This time $T_k$ corresponds to a communication update between nodes $i_k$ and $j_k$ started at time $T_k - \tau_{i_k j_k}$ [3]. Assumption 1 ensures that the communication started at time $T_k - \tau_{ij}$ takes some time $\tau^{(k)} \leqslant \tau_{i_k j_k}$ and is thus completed before time $T_k$ so that the update at time $T_k$ is indeed implementable. Consequently, the sequence $(x_t)_t$ generated by Algorithm 1 writes as:

$$
\begin{cases}
x_{T_k}(i) = x_{T_k-}(i) \quad \text{if} \quad i \notin \{i_k, j_k\}, \\[2mm]
x_{T_k}(i_k) \leftarrow x_{T_k-}(i_k) - \dfrac{K_{i_k j_k}}{p_{i_k j_k}} \big(x_{T_k - \tau_{i_k j_k}}(i_k) - x_{T_k - \tau_{i_k j_k}}(j_k)\big), \\[2mm]
x_{T_k}(j_k) \leftarrow x_{T_k-}(j_k) - \dfrac{K_{i_k j_k}}{p_{i_k j_k}} \big(x_{T_k - \tau_{i_k j_k}}(j_k) - x_{T_k - \tau_{i_k j_k}}(i_k)\big).
\end{cases}
$$

---

[3] Standard properties of P.p.p. guarantee that the sequence of points of $\mathcal{P}_{ij}$ translated by $\tau_{ij}$ is a P.p.p. with the same distribution.

---

**Algorithm 1** Delayed randomized gossip, edge $(ij)$

---

1: Step size $K_{ij} > 0$ and intensity $p_{ij} > 0$, Initialization $T_1(ij) \sim \text{Exp}(p_{ij})$
2: **for** $\ell = 1, 2, \ldots$ **do**
3:    $T_{\ell+1}(ij) = T_\ell(ij) + \text{Exp}(p_{ij})$.
4: **end for**
5: **for** $\ell = 1, 2, \ldots$ **do**
6:    At time $T_\ell(ij) - \tau_{ij}$ for, $i$ sends $\hat{x}_i = x_{T_\ell(ij)-\tau_{ij}}(i)$ to $j$ and $j$ sends $\hat{x}_j = x_{T_\ell(ij)-\tau_{ij}}(j)$ to $i$.
7:    At time $T_\ell(ij)$,

$$x_{T_\ell(ij)-}(i) \leftarrow x_{T_\ell(ij)-}(i) - \frac{K_{ij}}{p_{ij}}\big(\hat{x}_i - \hat{x}_j\big), \quad x_{T_\ell(ij)-}(j) \leftarrow x_{T_\ell(ij)-}(j) - \frac{K_{ij}}{p_{ij}}\big(\hat{x}_j - \hat{x}_i\big), \quad (8)$$

8: **end for**

---

Algorithm 1 is the pseudo-code for *Delayed Randomized Gossip*, from the viewpoint of two adjacent nodes $i$ and $j$. The times $T_\ell(ij)$ for $\ell \geqslant 1$ denote the activation times of edge $(ij)$. They follow a P.p.p. of intensity $p_{ij}$, and are sequentially determined by adjacent nodes $i$ and $j$. Formally, this decentralized and asynchronous algorithm corresponds to a jump process solution of a *delayed stochastic differential equation*. Defining $N(\text{d}t, (ij))$ as the *Poisson* measure on $\mathbb{R}^+ \times E$ of intensity $I\text{d}t \otimes \mathcal{U}_p$ where $\mathcal{U}_p$ is the probability distribution on $E$ proportional to $(p_{ij})_{(ij)\in E}$ $(\mathcal{U}_p((ij)) = p_{ij}/I)$, we have:

$$\text{d}x_t = -\int_{\mathbb{R}^+ \times E} \frac{K_{ij}}{p_{ij}} \nabla f_{ij}(x_{t-\tau_{ij}})\text{d}N(t, (ij)). \tag{7}$$

### 3.3 Convergence guarantees

**Theorem 1** (Delayed Randomized Gossip). *Assume the following bound on $K_{ij}$, $(ij) \in E$ holds, and let $\gamma > 0$ such that:[4]:*

$$K_{ij} \leqslant \frac{p_{ij}}{1 + \sum_{(kl)\sim(ij)} p_{kl}\big(\tau_{ij} + e\tau_{kl}\big)}, \quad \gamma \leqslant \min\left(\frac{\lambda_2\big(\Delta_G(\nu)\big)}{2}, \frac{1}{\tau_{\max}}\right), \tag{9}$$

*where $\nu_{ij} \equiv K_{ij}$, $(ij) \in E$, and $\tau_{\max} = \max_{(ij)\in E} \tau_{ij}$. For any $T \geqslant 0$, for $(x_t)_{t\geqslant 0}$ generated with delayed randomized gossip (Algorithm 1) or equivalently by the delayed SDE in Equation (7), we have:*

$$\mathbb{E}\left[\|\tilde{x}_T - \bar{x}\|^2\right] \leqslant e^{-\frac{\gamma T}{2}} \frac{1 + \frac{\tau_{\max}}{T}}{1 - \gamma\tau_{\max}}, \quad \tilde{x}_t = \gamma \frac{\int_0^t e^{\gamma s} x_s \text{d}s}{e^{\gamma t} - 1}. \tag{10}$$

An essential aspect of Theorem 1 lies in the explicit sufficient conditions for convergence it establishes for our proposed schemes, and on how they only rely on (upper bounds on) individual delays. We now discuss the ***asynchronous speedup*** obtained by fine-tuning algorithm parameters according to delays. For many graphs of interest such as grids, hypergrids, trees. . . and bounded edge parameters $\nu_{ij}$, in the large network limit $n \to \infty$ one has $\lambda_2(\Delta_G(\nu)) \to 0$[5] and so $\lambda_2(\Delta_G) \wedge 1/\tau_{\max} = \lambda_2(\Delta_G)$. The *asynchronous speedup* consists in having a rate of convergence as the eigengap of the Laplacian of the graph weighted by local communication constraints: the term $\lambda_2(\Delta_G(K))$, where each $K_{ij}$ is impacted only by local quantities. As mentioned in the introduction, this quantity should be understood as the analogue in decentralized optimization of the squared diameter of the graph (using time distances) in (3) in centrally coordinated algorithms and as expected, gossip algorithms are affected by spectral properties of the graph. In Theorem 1, these properties reflect delay heterogeneity across the graph: here, $\lambda_2(\Delta_G(K))^{-1}$ the mixing time of a random walk on the graph where jumping from node $i$ to $j$ takes a time $\tilde{\tau}_{ij} = K_{ij}^{-1}$. In contrast, previous analyses (of synchronous or asynchronous algorithms) involve the mixing time of a random walk with times between jumps set to a quantity that is linearly dependent on $\tau_{\max}$. We coin this discrepancy the *asynchronous speedup*.

Equation (9) suggests a scaling of $p_{ij} \approx 1/\tau_{ij}$, giving local weights $K_{ij}$ of order $1/(\text{degree}_{ij}\tau_{ij})$ where $\text{degree}_{ij}$ is the degree of edge $(ij)$ in the edge-edge graph. On the other hand, synchronous algorithms are slowed down by the slowest node: the equivalent term would be of order

---

[4]Note that $(ij) \sim (ij)$; constant $e$ is $\exp(1)$.
[5]Networks for which this fails are known as *expanders*.

$\lambda_2(\Delta_G(1/(\text{degree}_{ij}\tau_{\max})))$. Indeed, for a *gossip matrix* $W \in \mathbb{R}^{V \times V}$ ($W$ is a symmetric and stochastic matrix), the equivalent factor in synchronous gossip [9] is $\lambda_2(\Delta_G(W_{ij}\tau_{\max}))$, and $W_{ij}$ is usually set as $1/\text{degree}_{ij}$ in order to ensure convergence. Finally, assume that all $\tau_{ij}$ are equal to $\tau_{\max}$, and set $p_{ij} = 1/\tau_{\max}$. We then recover $\lambda_2(\Delta_G(1/(\text{degree}_{ij}\tau_{\max})))$ in the rate of convergence $\gamma$, thus yielding the same rates as synchronous algorithms [9] and asynchronous algorithms that only use a global upper bound on the delays [2, 23, 24, 34, 37, 39]: albeit being asynchronous, these algorithms do not take advantage of an asynchronous speedup in their convergence speed.

The same *asynchronous speedup* is present in the decentralized optimization problem, dealt with in Section 4, where we generalize our arguments using an augmented-graph approach [15]. This asynchronous speedup also holds under the presence of capacity constraints (Section 5), where we replace *P.p.p.*'s by more general *truncated P.p.p.*'s.

### 3.4 A delayed *ODE* for mean values in gossip

Before proving Theorem 1 in Appendix D, we provide some intuition for its conditions and the resulting convergence rate. We do this by studying the means of the iterates, that verify a delayed linear ordinary differential equation, easier to study than the process itself, for which we provide stability conditions. Denoting $y_t = \mathbb{E}[x_t] \in \mathbb{R}^{n \times d}$, for $t \geqslant 0$, where $(x_t)_{t \geqslant 0}$ is generated using delayed randomized gossip updates (6), we have:

$$\frac{\mathrm{d}y_t}{\mathrm{d}t} = -\sum_{(ij) \in E} K_{ij} \nabla f_{ij}(y_{t-\tau_{ij}}). \tag{11}$$

Indeed, for any $t \geqslant 0$ and $\mathrm{d}t > 0$,

$$\mathbb{E}[x_{t+\mathrm{d}t}|x_t] - x_t = -x_t + (1 - I\mathrm{d}t)x_t + o(\mathrm{d}t) + \mathrm{d}t \sum_{(ij) \in E} p_{ij}\big(x_t - \frac{K_{ij}}{p_{ij}} \nabla f_{ij}(x_{t-\tau_{ij}})\big),$$

and the right-handside simplifies as $-\mathrm{d}t \sum_{(ij) \in E} K_{ij} \nabla f_{ij}(x_{t-\tau_{ij}}) + o(\mathrm{d}t)$. Taking the mean, dividing by $\mathrm{d}t$ and making $\mathrm{d}t \to 0$ leads to the delayed ODE verified by $y_t = \mathbb{E}[x_t]$. Such delay-differential ODEs are classical [30] yet their stability properties are notoriously hard to characterize. This is typically attacked by means of *Lyapunov-Krasovskii* functionals or *Lyapunov-Razumikhin* functions [13]. Alternatively, sufficient conditions for convergence and stability guarantees on $(y_t)$ can be obtained, under specific conditions, by enforcing stability of the original system after *linearizing* it with respect to delays [26]. Linearizing in the sense of [26] means making the approximation $y_{t-\tau_{ij}} = y_t - \tau_{ij}\frac{\mathrm{d}y_t}{\mathrm{d}t}$. Under this approximation, we have:

$$\frac{\mathrm{d}y_t}{\mathrm{d}t} = -\sum_{(ij) \in E} K_{ij}\big(\nabla f_{ij}(y_t) - \tau_{ij}\nabla f_{ij}(\frac{\mathrm{d}y_t}{\mathrm{d}t})\big).$$

For any weights $\nu_{ij}$ and vector $z$, $\sum_{ij} \nu_{ij} \nabla f_{ij}(z) = \Delta_G(\nu)z$. Thus the delay-linearized ODE reads

$$(I - \Delta_G(\{K_{ij}\tau_{ij}\}))\frac{\mathrm{d}y_t}{\mathrm{d}t} = -\Delta_G(\{K_{ij}\})y_t. \tag{12}$$

This delay-linearized ODE (12) provides intuition on the behavior of $\mathbb{E}[x_t]$. Indeed, (12) is stable provided that $\rho(\Delta_G(\{\tau_{ij}K_{ij}\})) < 1$, in which case it has a linear rate of convergence of order $\lambda_2(\Delta_G(\{K_{ij}\}))$. Even though this stability condition and the rate of convergence are only heuristics, since (12) is obtained through an approximation of the delayed ODE verified by $\mathbb{E}[x_t]$ (11), this stability condition for the delay-linearized system implies stability of the original delayed system under assumptions on the matrices and delays involved[26], that hold in our case, leading to the following, proved in Appendix C.

**Proposition 1.** *Assume that the spectral radius of the weighted Laplacian $\Delta_G(\{\tau_{ij}K_{ij}\})$ verifies $\rho(\Delta_G(\{\tau_{ij}K_{ij}\})) < 1$. Then the delayed* ODE *(11) is stable.*

Consequently, the stability conditions (necessary conditions on step sizes $K_{ij}$ in Equation (9)) obtained in Theorem 1 are very natural. Indeed, a simple way to enforce $\rho(\Delta_G(\{\tau_{ij}K_{ij}\})) < 1$ based on local conditions consists in imposing $\sum_j \tau_{ij}K_{ij} < 1$ for all $i$. This is a weaker condition than the one stated in Theorem 1, but it only gives stability of the means. Furthermore, the rate of convergence of delayed randomized gossip in Theorem 1, that takes the form of the eigengap of a weighted graph Laplacian, is also that of any solution of the delay-linearized ODE (12).

# 4 Extension to decentralized optimization

In this section we extend our results to decentralized optimization, going beyond the quadratic objective functions considered for network averaging.

## 4.1 Delayed Decentralized Optimization

Consider the decentralized optimization problem (1). We make the following assumptions on the individual objective functions $f_i$ therein :

$$\text{Each } f_i, i \in V, \text{ is } \sigma\text{-strongly convex and } L\text{-smooth,} \tag{13}$$

see [6] for definitions. Let $f(z) := \sum_{i \in [n]} f_i(z)$ for $z \in \mathbb{R}^d$ and $F(x) = \sum_{i \in [n]} f_i(x_i)$ for $x = (x_1, \cdots, x_n) \in \mathbb{R}^{n \times d}$ where $x_i \in \mathbb{R}^d$ corresponds to node $i \in [n]$.

**Definition 2** (Fenchel Conjugate). *For any function $g : \mathbb{R}^p \to \mathbb{R}$, its* Fenchel conjugate *is denoted by $g^*$ and defined on $\mathbb{R}^p$ by $g^*(y) = \sup_{x \in \mathbb{R}^p} \langle x, y \rangle - g(x) \in \mathbb{R} \cup \{+\infty\}$.*

Our algorithm for delayed decentralized optimization is built on delayed randomized gossip for network averaging, augmented with local computations. Each node $i \in V$ keeps two local variables: the communication variable $x_i(t)$, used to run delayed randomized gossip, and a computation variable $y_i(t)$, used to make local computation updates in the following way.

**Local computations.** Each node $i$ generates a *Poisson point process* $\mathcal{P}_i^{\text{comp}} = \{T_1^{\text{comp}}(i) < T_2^{\text{comp}}(i), \ldots\}$ of intensity $p_i^{\text{comp}}$. At the clock tickings $T_k^{\text{comp}}(i)$, a local computation update is made corresponding to a computation started at a time $T_k^{\text{comp}}(i) - \tau_i^{\text{comp}}$, where $\tau_i^{\text{comp}}$ is the upper bound on the time to perform an elementary computation at node $i$, introduced in Assumption 1. Thus by assumption the computation started at time $T_k^{\text{comp}}(i) - \tau_i^{\text{comp}}$ is completed by time $T_k^{\text{comp}}(i)$ so that the update can be performed at that time. The precise form of this update is given by Equation (17).

**Communications.** In parallel of these local computations, a *Delayed Randomized Gossip* is run on the graph. Dedicated *P.p.p.* $(\mathcal{P}_{ij})_{(ij) \in E}$ with respective intensities $(p_{ij})_{(ij) \in E}$ are associated to communication updates of all network edges, and used to perform updates as prescribed by Equation (8) in *Delayed Randomized Gossip*.

The resulting Delayed Decentralized Optimization algorithm, or *DDO* for short, is described in Algorithm 3 and is a combination of Algorithm 1 for communication updates along edges $(ij) \in E$ with Algorithm 2 for local computation updates at nodes $i \in V$.

## 4.2 Convergence guarantees

The process $(x(t), y(t)) \in \mathbb{R}^{2n \times d}$ defined by algorithm *DDO*, Algorithm 3, satisfies the following convergence guarantees that generalize Theorem 1 to decentralized optimization beyond the case of quadratic functions.

**Theorem 2** (Delayed Decentralized Optimization). *Under the regularity assumptions* (13)*, assume further that for all $1 \leqslant i \in V$ and $(ij) \in E$, we have:*

$$
\begin{aligned}
K_{ij} &\leqslant \frac{p_{ij}}{1 + \sum_{(kl) \sim (ij)} p_{kl}\left(\tau_{ij} + e\tau_{kl}\right)} \\
K_i^{\text{comp}} &\leqslant \frac{p_i^{\text{comp}}}{1 + \sum_{j \sim i} p_{ij}\left(\tau_i^{\text{comp}} + e\tau_{ij}\right)} \,.
\end{aligned}
\tag{14}
$$

*Let $\tau_{\max} := \max\left(\max_{(ij) \in E} \tau_{ij}, \max_{i \in V} \tau_i^{\text{comp}}\right)$. Then for $\gamma > 0$ such that*

$$\gamma \leqslant \min\left(\frac{\sigma}{4L}\lambda_2\big(\Delta_G(K)\big), \frac{1}{\tau_{\max}}\right), \tag{15}$$

*the process $(x(t), y(t))$ generated by* DDO *satisfy*

$$\frac{\int_0^T e^{\gamma t} \mathbb{E}\left[\left\|\frac{\sigma}{2}x(t) - \bar{x}^\star\right\|^2\right] \mathrm{d}t}{\int_0^T e^{\gamma t}\left\|\frac{\sigma}{2}x(0) - \bar{x}^\star\right\|^2 \mathrm{d}t} \leqslant e^{-\frac{\gamma T}{2}} \frac{L}{\sigma} \frac{1 + \frac{\tau_{\max}}{T}}{1 - \gamma\tau_{\max}}, \tag{16}$$

*where* $\bar{x}^\star = (x^\star, \ldots, x^\star)^\top \in \mathbb{R}^{n \times d}$ *for* $x^\star$ *minimizer of* $f = \sum_i f_i$.

DDO is based on a dual formulation and uses an augmented graph representation introduced in [16] to decouple computations from communications, as detailed in the proof. The dual gradient computations in Algorithm 2 can be expensive in general; they could be avoided by using a primal-dual approach for the computation updates [21].

The convergence guarantees we obtain resemble classical ones: Interpreting $\gamma$ as the reciprocal of the time scale for convergence, we recognize in its upper bound (15) an "optimization factor" $\kappa_{\text{comp}}^{-1} := \sigma/L$, and a "communication factor" $\kappa_{\text{comm}}^{-1} = \lambda_2\big(\Delta_G(K)\big)$. Our method is non-accelerated, so the computation factor $\kappa_{\text{comp}}$, the condition number of the optimization problem, is expected. The communication factor captures the delay heterogeneity in the graph as in *Delayed Randomized Gossip*, leading to the *asynchronous speedup* discussed in Section 3 after Theorem 1.

Previous approaches have considered accelerating decentralized optimization by obtaining $\sqrt{\kappa_{\text{comp}}}$ instead of $\kappa_{\text{comp}}$ and/or $\sqrt{\kappa'_{\text{comm}}}$ instead of $\kappa'_{\text{comm}}$ for $\kappa'_{\text{comm}}$ a communication factor in the rate of convergence [15, 20, 32]. Our result yields a speedup of a different nature: we obtain a communication factor $\kappa_{\text{comm}}$ that can be arbitrarily larger than previously considered $\kappa'_{\text{comm}}$ for networks with huge delay heterogeneity.

---

**Algorithm 2** Local computations, node $i$

---

1: Step size $K_i^{\text{comp}} > 0$
2: Initialization $x_0(i) = y_0(i) = 0$
3: Initialization $T_1^{\text{comp}}(i) \sim \text{Exp}(p_i^{\text{comp}})$
4: **for** $\ell = 1, 2, \ldots$ **do**
5: $\quad T_{\ell+1}(ij) = T_\ell(ij) + \text{Exp}(p_{ij})$.
6: **end for**
7: **for** $\ell = 1, 2, \ldots$ **do**
8: $\quad$ At time $T_\ell^{\text{comp}}(i) - \tau_i^{\text{comp}}$, node $i$ computes $g_i = \nabla\phi_i^*(y_i(T_\ell^{\text{comp}}(i) - \tau_i^{\text{comp}}))$ (takes a time less than $\tau_i^{\text{comp}}$) and keeps $\hat{x}_i = x_i(T_\ell^{\text{comp}}(i) - \tau_i^{\text{comp}})$ in memory, where $\phi_i = f_i - \frac{\sigma}{4}\|.\|^2$.
9: $\quad$ At time $T_\ell^{\text{comp}}(i)$,

$$
\begin{aligned}
y_i(T_\ell^{\text{comp}}(i)) &\xleftarrow{t} y_i(T_\ell^{\text{comp}}(i)-) - \frac{\sigma K_i^{\text{comp}}}{p_i^{\text{comp}}}\big(g_i - \hat{x}_i\big)\,, \\
x_i(T_\ell^{\text{comp}}(i)) &\xleftarrow{t} x_i(T_\ell^{\text{comp}}(i)-) - \frac{K_i^{\text{comp}}}{2p_i^{\text{comp}}}\big(\hat{x}_i - g_i\big)\,.
\end{aligned}
\tag{17}
$$

10: **end for**

---

**Algorithm 3** *DDO*

---

1: Node initializations $x_0(i) = y_0(i) = 0$, $i = 1, \ldots, n$
2: **for** $i \in V$ and $(ij) \in E$, asynchronously, in parallel **do**
3: $\quad$ Communication updates along edge $(ij)$ according to Algorithm 1
4: $\quad$ Local computation updates at node $i$ according to Algorithm 2
5: **end for**
6: **Output:** $\frac{\sigma}{2}x_i(t)$ at time $t$ and node $i$.

---

## 5 Handling communication and computation capacity limits

### 5.1 Communication and computation capacity constraints

A given node or edge in the network may be able to handle only a limited number of communications or computations simultaneously. In Delayed Randomized Gossip and *DDO* algorithms, such con-

straints could be violated when some P.p.p. generates many points in a short interval. We extend our algorithms and resulting convergence guarantees to take into account these additional constraints.

In the continuized framework, this constraint can be enforced by truncating the *P.p.p.* that handles activations (Definition 3). We formalize communication and capacity constraints in Assumption 2, and show that asynchronous speedup is still achieved in this setting in Theorem 3.

In the previous sections, step size parameters $K_{ij}, K_i^{\text{comp}}$ of the algorithms could be tuned to counterweight the effect of delays for arbitrary intensities $p_{ij}$. With the introduction of capacity constraints we will see that the local optimizers at every node must also bound the intensities $p_{ij}, p_i^{\text{comp}}$ based on local quantities. The resulting rate of convergence is the same as in Theorems 1 and 2, up to a constant factor of $1/2$.

We formalize communication and computation capacity constraints as follows.

**Assumption 2** (Capacity constraints). *For some $q_{ij}, q_i^{\text{comm}}, q_i^{\text{comp}} \in \mathbb{N}^* \cup \{\infty\}$, $i \in V$ and $(ij) \in E$,*

1. Computation Capacity: *Node $i$ can compute only $q_i^{\text{comp}}$ gradients in an interval of time of length $\tau_i^{\text{comp}}$;*

2. Communication Capacity, edge-wise limitations: *Only $q_{ij}$ messages can be exchanged simultaneously between adjacent nodes $i \sim j$ in an interval of time of length $\tau_{ij}$;*

3. Communication Capacity, node-wise limitations: *Node $i$ can only send $q_i^{\text{comm}}$ messages in any interval of time of length $\tau_i^{\text{comm}} = \max_{j \sim i} \tau_{ij}$.*

Taking into account these constraints in the analysis boils down to replacing *P.p.p.* processes $(\mathcal{P}_{ij})_{(ij) \in E}$, $(\mathcal{P}_i^{\text{comp}})_{i \in V}$ of intensities $(p_{ij})$, $(p_i^{\text{comp}})$ in the *DDO* algorithm, by *truncated Poisson point processes* $(\tilde{\mathcal{P}}_{ij}, \tilde{\mathcal{P}}_i^{\text{comp}})$ (see Definition 3).

More precisely, for every edge $(ij) \in E$ (*resp.* node $i \in V$), let $n_{ij}(t)$ be the number of communications occurring along $(ij)$ between times $t - \tau_{ij}$ and $t$ (*resp.* $n_{i,j}^{\text{comm}}$ the number of communications node $i$ is involved in between times $t$ and $t - \tau_{ij}$, $n_i^{\text{comp}}$ the number of computations node $i$ is involved with between times $t$ and $t - \tau_i^{\text{comp}}$). Without capacity constraints, these quantities are discrete Poisson random variables (of mean $p_{ij}\tau_{ij}$ for $n_{ij}(t)$, *e.g.*).

## 5.2 Convergence guarantees

As in Section 4, we consider communication and computation update rules as in Algorithm 3 (*DDO* algorithm). In the presence of capacity constraints, a communication alongside edge $(ij) \in E$ at a clock ticking $t \in \mathcal{P}_{ij}$ occurs and *does not break the communication capacity constraints* if and only if $n_{ij}(t) < q_{ij}$ (for edge-wise limitations), $n_{i,j}^{\text{comm}}(t) < q_i^{\text{comm}}$ and $n_{j,i}^{\text{comm}}(t) < q_j^{\text{comm}}$ (for node-wise limitations) are satisfied. The realistic implementation of these *truncated Point processes* is discussed in Section 7.

Under capacity constraints, we have the following guarantees for our algorithm, defined as in Algorithm 3 (Algorithm 1 for communications and Algorithm 2 for local computations), where communications and computations that violate the capacity constraints are dropped.

**Theorem 3.** *Assume for any $i \in V$ and $(ij) \in E$:*

$$
\begin{aligned}
cp_i^{\text{comp}}\tau_i^{\text{comp}} &\leqslant q_i^{\text{comp}}, \\
cp_{ij}\tau_{ij} &\leqslant q_{ij}, \\
c\sum_{j \sim i} p_{ij}\tau_i^{\text{comm}} &\leqslant q_i^{\text{comm}},
\end{aligned}
\tag{18}
$$

*where $c = 1/(1 - \sqrt{\ln(6)/2})$ is a numerical constant. Then, if the assumptions of Theorem 2 described in Equation 14 additionally hold, for $\gamma$ verifying*

$$
\gamma \leqslant \min\left( \frac{\sigma}{8L}\lambda_2\big(\Delta_G(\nu_{ij} = K_{ij})\big), \frac{1}{\tau_{\max}} \right),
$$

*we have:*

$$
\frac{\int_0^T e^{\gamma t}\mathbb{E}\left[\left\| \frac{\sigma}{2}x(t) - \bar{x}^\star \right\|^2\right]\mathrm{d}t}{\int_0^T e^{\gamma t}\left\| \frac{\sigma}{2}x(0) - \bar{x}^\star \right\|^2 \mathrm{d}t} \leqslant e^{-\frac{\gamma T}{2}} \frac{L}{\sigma} \frac{1 + \frac{\tau_{\max}}{T}}{1 - \gamma\tau_{\max}}.
$$

The same guarantees as without the capacity constraints thus hold, up to a constant factor $1/2$ in the rate of convergence. The conditions on the activation intensities (18) suggest that graph sparsity is beneficial: for $q_i^{\text{comm}}$ small, $2\sum_{j\sim i} p_{ij}\tau_i^{\text{comm}} \leqslant q_i^{\text{comm}}$ translates into $p_{ij}$ scaling with the inverse of the edge-degree of $(ij)$, so large degrees thus slow down the convergence. The new conditions (18) are easily enforced with the natural choice of intensities $p_{ij}$ (*resp.* $p_i^{\text{comp}}$) of order $1/\tau_{ij}$ (*resp.* $\tau_i^{\text{comp}}$).

Taking $q_i^{\text{comm}} = 1$, we recover the behavior of *loss networks* [17], where a node cannot concurrently communicate with different neighbors. Gossip on loss networks was previously studied in [11], to obtain some form of asynchronous speedup. Comparatively, our present algorithms are structurally simpler and their analysis in the continuized framework yields faster convergence speeds.

## 6    Braess's Paradox and Experiments

In this section, we investigate how the local step sizes $K_{ij}$ and Poisson intensities $p_{ij}$ used in Theorems 1, 2 and 3 should be tuned for a fixed choice of communication delays. Consider the line graph with constant delays $\tau_{i,i+1} = \tau$. Add edge $(1, n)$ in order to close the line, with a delay $\tau_{1n} = \tau'$ with arbitrarily large $\tau'$. If the added Poisson intensity $p_{1n}$ satisfies $\tau_{1n}p_{1n} \to \infty$, then according to Theorem 1, we have $K_{12} \to 0$ and $K_{n-1,n} \to 0$. Consequently, since $\gamma = \mathcal{O}(\Delta_G(K))$, we have $\gamma \to 0$: the weighted graph becomes close to disconnected. By adding an edge to the graph, the convergence speed of delayed randomized gossip is degraded.

In order to alleviate the phenomenon, we would need to virtually delete the edge, by setting $p_{1n} = 0$. Figure 1 illustrates this phenomenon in the more general setting: one can sparsify the communication graph by solving a regularized optimization problem over the $p_{ij}$ in order to maximize $\lambda_2(\Delta_G(K))$ ($K$ being a function of $p$), leading to both faster consensus and smaller communication complexity (and thus lower energy footprint).

In road-traffic, removing one or more roads in a road network can speed up the overall traffic flow. This phenomenon, called *Braess's paradox* [10], also arises in loss networks [3]. In our problem, this translates to removing an edge $(ij)$ with a non-negligible Poisson intensity $p_{ij}$. We take $G_1$ a dense Erdős-Rényi random graph (Figure 1(a)) of parameters $n = 30, p = 0.75$. Delays $\tau_{ij}$ are taken equal to 0.01 with probability 0.9, and to 1 with probability 0.1. Initially, intensities are set as $p_{ij}^{(1)} = 1/\tau_{ij}$. Maximizing:

$$\lambda_2\left(\Delta_G\left(\frac{p_{ij}}{1 + \sum_{kl\sim ij} p_{kl}(\tau_{ij} + e\tau_{kl})}\right)\right) - \omega\sum_{(ij)\in E} p_{ij}\tau_{ij}$$

over $(p_{ij})_{ij}$, we obtain intensities $p^{(2)}$ and a graph $G_2$ (Figure 1(b)), sparser than $G_1$: we delete edges that have a null intensity (*i.e.* such that $p_{ij}^{(2)} = 0$). We then run our delayed gossip algorithm for initialization $x_0$ a Dirac mass ($x_0(i) = I_{i=i_0}$), on $G_1$ (blue curves) and $G_2$, for the choice of $K_{ij}$ as in Theorem 1. The green curve is the synchronous gossip algorithm [9] on $G_1$, to illustrate the asynchronous speedup, where each iteration takes a time $\tau_{\max} = 1$. In Figure 1(c), the error to the consensus is measured as a function of the continuous time, while it is measured in terms of number of updates in Figure 1(d) and in terms of energy (defined as $\sum_{k:T_k<t}\tau_{i_kj_k}$ at time $t$: the energy consumed by a communication is assumed to be proportional to the time the communication took) in Figure 1(e).

As expected, in terms of number of updates in the whole graph and energy spent, the sparser graph is more effective: slow and costly edges were deleted. Perhaps more surprising, but supported by our theory (Theorem 2) and the resulting Braess's paradox, this also holds in Figure 1(c): even though in the same amount of time, less updates are made in the sparser graph $G_2$ than in $G_1$, delayed randomized gossip is still faster on $G_2$ than $G_1$. Making less updates and deleting some communications make all other communications more efficient.

We believe that this phenomenon could be exploited for efficient design of large scale networks, beyond the maximization the spectral gap regardless of physical constraints as in [41] for instance.

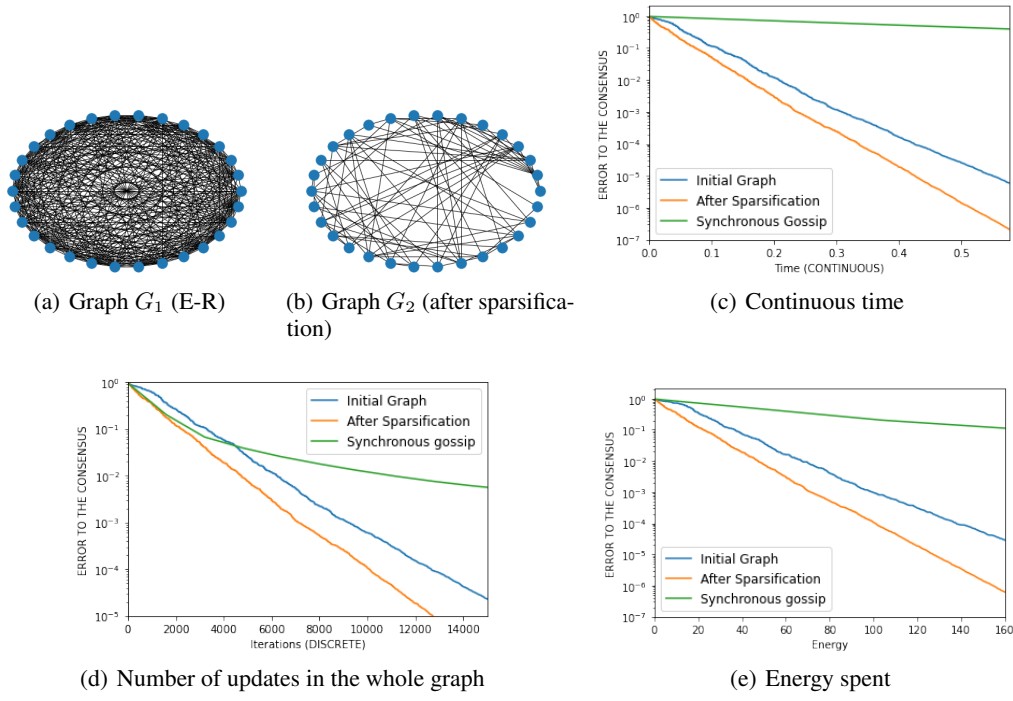

(a) Graph $G_1$ (E-R)

(b) Graph $G_2$ (after sparsification)

(c) Continuous time

(d) Number of updates in the whole graph

(e) Energy spent

Figure 1: **Experiments and Braess's paradox.**

## 7 Decentralized generation of *truncated Poison point processes*

### 7.1 Practical Implementation and Limitations of Algorithm 3 (*DDO*)

In order for *DDO* (or Delayed Randomized Gossip) to be implemented (with or without *truncated P.p.p.*), some conditions are required, leading to some limitations that we discuss below. Communication steps at some time $t$ are of the form $-\eta_{ij}(x_i(t - \tau_{ij}) - x_j(t - \tau_{ij}))$ (on node $i$): the (delayed) iterates of local variables $x_i$ and $x_j$ for this update are required to have the exact same delay. This comes from our dual formulation. Secondly, it is not clear who has the initiative of communications in our algorithm, since we treat delays and communications edge-wise. However, we propose here a practical implementation of *DDO*, taking into account all these limitations: nodes have the initiative of communications and delayed variables are enforced to have the same constant delays (equal to an upper-bound on these delays).

**Computation Updates at node** $i$    let $\mathcal{N}_i^{\text{comp}}(t)$ be the number of computations launched at node $i$ between times $t$ and $t - \tau_i^{\text{comp}}$. Initialize $\mathcal{P}_i^{\text{comp}}$ as $\{t_0\}$ where $t_0$ is a random time of exponential law of parameter $p_i^{\text{comp}}$. At any time $t \in \mathcal{P}_i^{\text{comp}}$:

1. Compute a random time $T$ of exponential law of parameter $p_i^{\text{comp}}$, and add $t + T$ to $\mathcal{P}_i^{\text{comp}}$;

2. *Computation:* at time $t$, if $\mathcal{N}_i^{\text{comp}}(t) < q_i^{\text{comp}}$, $i$ computes $\nabla g_{i^{\text{comp}}}^*(y_i^{\text{comp}}(t))$ and saves into memory node-variable $x_i(t)$;

3. *Update:* at time $t + \tau_i^{\text{comp}}$, the computation is finished, and the computation update (17) can be performed.

**Communication Updates at node** $i$    let $\mathcal{N}_i^{\text{comm}}(t)$ and $\mathcal{N}_{ij}(t)$ for $j \sim i$ respectively be the number of communications started by node $i$ between times $t$ and $t - \tau_i^{\text{comm}}$ and the number of communications started between nodes $i$ and $j$ in $[t - \tau_{ij}, t]$. Initialize $\mathcal{P}_i^{\text{comm}}$ as $\{t_0\}$ where $t_0$ is a random time of exponential law of parameter $\sum_{j \sim i} p_{ij}/2$. At any time $t \in \mathcal{P}_i^{\text{comm}}$:

1. Compute a random time $T$ of exponential law of parameter $\sum_{j \sim i} p_{ij}/2$, and add $t + T$ to $\mathcal{P}_i^{\text{comm}}$;

2. *Local synchronization:* at time $t$, if $\mathcal{N}_i^{\mathrm{comm}}(t) < q_i^{\mathrm{comm}}$, $i$ chooses an adjacent node $j$ with probability $p_{ij}/\sum_{k\sim i} p_{ik}$, and sends a *ping* (smallest message possible). We assume that sending a *ping* takes a time upper-bounded by $\tau_{ij}^{\mathrm{ping}}$. Upon reception of this *ping*, if $\mathcal{N}_j^{\mathrm{comm}}(t) < q_j^{\mathrm{comm}}$, $j$ returns the same *ping* to $i$. $i$ and $j$ are thus synchronized at time $t + 2\tau_{ij}^{\mathrm{ping}}$. Until the return of the feedback by node $j$, $i$ assumes in its request list that this communication will happen.

3. *Communication:* if $\mathcal{N}_{ij}(t + 2\tau_{ij}^{\mathrm{ping}}) < q_{ij}$, $i$ sends $x_i(t)$ to $j$ and $j$ sends $x_j(t)$ to $i$, while each agent keeps in memory the vector sent. For this to be possible, at a time $s$, node $i$ needs to keep in memory its local values at times between $s - \max_{j\sim i} \tau_{ij}^{\mathrm{ping}}$ and $s$ (a small number of values usually, so that is not too restrictive).

4. *Update:* at time $t + 2\tau_{ij}^{\mathrm{ping}} + \tau_{ij}$, update (8) can thus be performed.

The induced process has the same law as the one studied and analyzed thanks to the classical property that a P.p.p. with intensity $p$ has its distribution unchanged after translation of all its points by some constant $\tau_{ij}$. The communication/computation scheme above emphasizes the fact that quantities $\tau_{ij}, \tau_i^{\mathrm{comp}}$ are upper bounds on the delays of local communication/computations. The delay is here $\tau_{ij} + 2\tau_{ij}^{\mathrm{ping}}$ instead of simply $\tau_{ij}$ yet these are of the same order since $\tau_{ij}^{\mathrm{ping}} \leqslant \tau_{ij}$.

## Conclusion

We introduced a novel analysis framework for the study of asynchronous algorithms in the presence of delays, establishing that an *asynchronous speedup* can be achieved in the network averaging problem, and in decentralized optimization. Our results hold for explicit choices of algorithm parameters based on local network characteristics. They derive from the continuous-time analysis and assumptions handled in our continuized framework. The explicit conditions and convergence rates we obtain allow us to further discuss counter-intuitive effects akin to the Braess paradox, such as the possibility to speed up convergence by suppressing communication links.

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

# A  Discussion of our assumptions

**Continuous time clocks**  Implicitly we have assumed that agents in the networks share a continuous-time clock.

We now discuss how critical this assumption is. One may wonder if clock skews, drifts or shifts between agents could lead to unwanted behavior (instability). First, computer clocks are synchronized via Network Time Protocol (NTP) or other more recent and more robust protocols, justifying the assumption of a common shared clock. However, failures in these protocols could lead to slight skews, that need to be addressed. Looking at our algorithm, it in fact appears that, using timestamps, nodes do not need to be perfectly synchronized, as long as time in each local clock passes at the same speed. Shifts in time-synchronization thus do not appear to be the core practical difficulty: clock drifts (clocks that do not have the same frequency and where time thus passes at a different speed) are the remaining issue. However, this can be dealt with simply by augmenting pairwise delays to take that into account.

**Local upper bounds of the local delays**  Our rates depend on local upper bounds of the local delays, that need to be known in order to tune the coefficients. The drawback is thus that, if a node/edge has an erratic behavior (fast but rarely slow), we only use the upper-bound on the delays, which can lead to slower convergence. However, we believe that this is a drawback of most (if not all) asynchronous algorithms, where delays are dealt with by diminishing step sizes. These step sizes are tuned using upper bounds on the delays. Even if we did not artificially force the delays to be equal to the local upper bound $\tau_{ij}$, we would have to tune $K_{ij}$ based on these upper bounds, resulting in the same rates of convergence. A natural extension would be to consider whether step sizes can be adapted to the physical delay as in [27] for asynchronous SGD, therefore obtaining an asynchronous speedup and guarantees without requiring any knowledge on the delay upper bounds.

# B  Delayed coordinate gradient descent in the continuized framework

Let $G$ be a $\sigma$-strongly convex function on $\mathbb{R}^D$. For $k = 1, ..., m$, let $E_k$ be a subspace of $\mathbb{R}^d$, and assume that:

$$\mathbb{R}^d = \bigoplus_{k=1}^{n} E_k \,. \tag{19}$$

For $x \in \mathbb{R}^D$, let $x_k$ denote its orthogonal projection on $E_k$ and let $\nabla_k G := (\nabla G)_k$, and assume that the subspaces $E_1, ..., E_m$ are orthogonal. For $k, \ell \in [m]$, we say that $k$ and $\ell$ are adjacent and we write $k \sim \ell$ if and only if $\nabla_k \nabla_\ell G$ is not identically constant equal to $0$. This induces a symmetric graph structure on the coordinates $k \in [m]$. In the context of gossip network averaging, $m = |E|$ and each subspace $E_k$ corresponds to an edge $e_k = (i_k j_k)$ of the graph; in that context, we have $k \sim \ell$ if and only if edges $e_k$ and $e_\ell$ share a node.

In the network averaging problem previously described, the function $G$ used is $g(\lambda) = \frac{1}{2}\|A\lambda\|$ for $\lambda \in \mathbb{R}^{E \times d}$ the edge variables. Subspaces are $E_{ij}$ of dimension $d$ for $(ij) \in E$ (and $m = |E|$) corresponding to variables of $\lambda$ associated to edge $(ij)$.

## B.1  Algorithm and assumptions

### B.1.1  Continuized delayed coordinate gradient descent algorithm

For $k \in [m]$, let $\mathcal{P}_k$ be a *P.p.p.* of intensity $p_k$ denoting the times at which an update can be performed on subspace $E_k$. For $t \in \mathcal{P}_k$ let $\varepsilon_k(t) \in \{0, 1\}$ be the indicator of whether the update is performed or not. Let also $\eta_k$ be some positive step size for $k \in [m]$. Consider then the following continuous-time process $X(t)$, where $X_k(t)$ (the projection of $X(t)$ on $E_k$) evolves according to:

$$\mathrm{d}X_k(t) = -\varepsilon_k(t)\eta_k \nabla_k G((X(t - \tau_k))\mathcal{P}_k(\mathrm{d}t)\,, \tag{20}$$

where $\mathcal{P}_k(\mathrm{d}t)$ corresponds to a Dirac at the points of the *P.p.p.* $\mathcal{P}_k$. In words, $(X(t))_{t \geqslant 0}$ is a jump process that takes coordinate gradient descent steps along subspaces $(E_k)_{k \in [m]}$ at the times of independent Poisson point processes $(\mathcal{P}_k)_{k \in [m]}$. We introduced variables $(\varepsilon_k(t))_{k \in [m], t \in \mathcal{P}_k}$ with values in $\{0, 1\}$ to represent capacity constraints: $\varepsilon_k(t) = 0$ if the update at time $t \in \mathcal{P}_k$ cannot be

performed due to some constraint saturation; these variables $\varepsilon_k(t)$ will be essential in our treatment of communication and computation capacity constraints in Section 5.

### B.1.2 Regularity assumptions

$G$ is $\sigma$-strongly convex, and $L_k$-smooth on $E_k$ for $k \in [m]$. Furthermore, there exist non-negative real numbers $M_{k,\ell}$ and $M_{\ell,k}$ for $k \sim \ell$ such that for all $k = 1, ..., m$ and $x, y \in \mathbb{R}^D$, we have:

$$\|\nabla_k G(x) - \nabla_k G(y)\| \leqslant \sum_{\ell \sim k} M_{k,\ell}\|x_\ell - y_\ell\| \, . \tag{21}$$

When $G$ is $L_k$ smooth on $E_k$ as we assume, the above condition is verified by the choice $M_{i,j} = L_j$, $i \sim j$. If $\nabla_k G$ is $M_k$-Lipschitz, Condition (21) is verified by the choice $M_{k,\ell} = M_k$. Assumption (21) however allows for more freedom, and is particularly well suited for our analysis. In particular for decentralized optimization, it will be convenient to take $M_{k\ell} = \sqrt{L_k L_\ell}$.

### B.1.3 Assumptions on variables $\varepsilon_k(t)$, $t \in \mathcal{P}_k$

For $t \in \mathcal{P}_k$, random variable $\varepsilon_k(t)$ is $\sigma\big(\mathcal{P}_\ell \cap [t - \tau_k, t), \ell \in [m]\big)$-measurable, and there exists a constant $\varepsilon_k > 0$ such that:
$$\mathbb{E}\left[\varepsilon_k(t)\right] \geqslant \varepsilon_k \, ,$$
Furthermore, we assume that $\varepsilon_k(t)$ is negatively correlated with each quantity $N_\ell(t - \tau_k, t) = |\mathcal{P}_\ell \cap [t - \tau_k, t]|$, *i.e.* that for all $k, \ell \in [m]$,

$$\mathbb{E}\left[\varepsilon_k(t) N_\ell(t - \tau_k, t)\right] \leqslant \mathbb{E}\left[\varepsilon_k(t)\right]\mathbb{E}\left[N_\ell(t - \tau_k, t)\right] \, . \tag{22}$$

In our subsequent treatment of communication and capacity contraints, we shall see that the above assumptions are verified for $\epsilon_k(t)$ the indicator that $t$ is a point a *truncated P.p.p.* $\tilde{\mathcal{P}}_k$ defined as follows:

**Definition 3 (Truncated *P.p.p.*).** *Let $(\mathcal{P}_k)_{1 \leqslant k \leqslant m}$ be P.p.p. of respective intensities $(p_k)_{1 \leqslant k \leqslant m}$, $(\tau_k)_{1 \leqslant k \leqslant m}$ non-negative delays. Let $N_k$ be the Poisson point measures associated to $\mathcal{P}_k$, $k \in [m]$. For $(\mathcal{C}_r)_{1 \leqslant r \leqslant M}$ subsets of $[m]$, we define the truncated Poisson point measures $(\tilde{N})_{1 \leqslant k \leqslant m}$ of intensities $(p_k)_{1 \leqslant k \leqslant m}$ and parameters $(\tau_k)_k, (q_{k,r})_{k \in [m], r \in [M]}$ as:*

$$\mathrm{d}\tilde{N}_k(t) = 1_{\left\{\bigcap_{1 \leqslant k \leqslant M}\left\{\sum_{\ell \in \mathcal{C}_r} N_\ell([t - \tau_k, t)) \leqslant q_{k,r}\right\}\right\}}\mathrm{d}N_k(t) \, , \tag{23}$$

*and we let $\tilde{\mathcal{P}}_k$ be the point process associated to this point measure.*

## B.2 Convergence guarantees and analysis

The main result of this Section is the following

**Theorem 4 (Delayed Coordinate Gradient Descent).** *Under the stated assumptions on regularity of $G$ and on variables $\epsilon_k(t)$, assume further that the step sizes $\eta_k$ are given by $\eta_k = \frac{K_k}{p_k L_k}$ where for all $k \in [m]$,*

$$K_k \leqslant \frac{p_k}{1 + \sum_{\ell \sim k} p_\ell \left(\frac{\tau_k M_{k,\ell} + e\tau_\ell M_{\ell,k}}{\sqrt{L_k L_\ell}}\right)} \, , \tag{24}$$

*and let $\gamma \in \mathbb{R}_+$ be such that:*

$$\gamma < \min\left(\sigma \min_k \frac{\varepsilon_k K_k}{L_k}, \frac{1}{\tau_{\max}}\right) \, , \tag{25}$$

*where $\tau_{\max} := \max_{k \in [m]} \tau_k$. Then for any $T > 0$ the solution $X(t)$ to Equation (20) verifies*

$$\frac{\int_0^T e^{\gamma t}\mathbb{E}\left[G(X(t)) - G(x^\star)\right]\mathrm{d}t}{\int_0^T e^{\gamma t}\big(G(X(0)) - G(x^\star)\big)\mathrm{d}t} \leqslant e^{-\frac{\gamma T}{2}}\frac{1 + \frac{\tau_{\max}}{T}}{1 - \gamma \tau_{\max}} \, . \tag{26}$$

*Proof.* We proceed in three steps. The first step consists in upper bounding, for $t \geqslant 0$, the quantity $\frac{\mathrm{d}\mathbb{E}[G(X(t))]}{\mathrm{d}t}$. We then introduce in Step 2 a Lyapunov function inspired by the Lyapunov-Krasovskii functional [13]), and by using the result proved in the first step, we show that it verifies a delayed ordinary differential inequality. The last step then consists in deriving the desired result from this delayed differential inequality.

**Step 1** To bound $\frac{\mathrm{d}\mathbb{E}[G(X(t))]}{\mathrm{d}t}$, we study infinitesimal increments between $t$ and $t + \mathrm{d}t$ for $\mathrm{d}t \to 0$. This approach is justified by results on *stochastic ordinary differential equation with Poisson jumps*, see [8]. For $t \geqslant 0$, let $\mathcal{F}_t$ be the filtration induced by $\mathcal{P}_k \cap [0,t)$, $k \in [m]$ *i.e.*, the filtration up to time $t$. By convention, for non-positive $t$, we write $X(t) = X(0)$. The following inequalities are written up to $o(\mathrm{d}t)$ terms, that we omit to lighten notations. Finally, we write

$$g_{k,t} = \nabla_k G(X(t)), \quad k \in [m], t \geqslant 0.$$

We have, using local smoothness properties of $G$ and the fact that for a *P.p.p.* $\mathcal{P}$ of intensity $p$, $\mathbb{P}(\mathcal{P} \cap [t, t + \mathrm{d}t] = \emptyset) = 1 - p\mathrm{d}t + o(\mathrm{d}t)$ and $\mathbb{P}(\#\mathcal{P} \cap [t, t + \mathrm{d}t] = 1) = p\mathrm{d}t + o(\mathrm{d}t)$:

$$\frac{\mathbb{E}\left[G(X(t+\mathrm{d}t)) - G(X(t))|\mathcal{F}_t\right]}{\mathrm{d}t}$$

$$= \sum_{k=1}^{m} p_k \left( G\left(X(t) - \frac{\varepsilon_k(t)K_k}{p_k L_k} g_{k,t-\tau_k}\right) - G(X(t)) \right)$$

$$\leqslant \sum_{k=1}^{m} p_k \left( -\frac{K_k}{p_k L_k} \langle \varepsilon_k(t) g_{k,t-\tau_k}, g_{k,t} \rangle + \frac{L_k}{2} \left\| \varepsilon_k(t) \frac{K_k}{p_k L_k} \nabla_k g_{k,t-\tau_k} \right\|^2 \right).$$

First, we rewrite $-\frac{\varepsilon_k(t)K_k}{p_k L_k}\langle g_{k,t-\tau_k}, g_{k,t}\rangle$ as

$$-\frac{\varepsilon_k(t)K_k}{p_k L_k}\|g_{k,t-\tau_k}\|^2 - \frac{\varepsilon_k(t)K_k}{p_k L_k}\langle g_{k,t-\tau_k}, g_{k,t} - g_{k,t-\tau_k}\rangle,$$

and bound the second term there by

$$-\frac{\varepsilon_k(t)K_k}{p_k L_k}\langle g_{k,t-\tau_k}, g_{k,t} - g_{k,t-\tau_k}\rangle$$

$$\leqslant \frac{\varepsilon_k(t)K_k}{p_k L_k}\|g_{k,t-\tau_k}\| \|g_{k,t} - g_{k,t-\tau_k}\|$$

$$\leqslant \frac{K_k}{p_k L_k}\|\varepsilon_k(t)g_{k,t-\tau_k}\| \sum_{\ell \sim k} M_{k,\ell} \|X_\ell(t) - X_\ell(t-\tau_k)\|,$$

where we used the Cauchy-Schwarz inequality and then local Lipschitz property (21) of $\nabla_k G$. Writing

$$\|X_\ell(t) - X_\ell(t-\tau_k)\|$$

$$= \left\| \int_{(t-\tau_k)^+}^{t} \frac{\varepsilon_\ell(s)K_\ell}{p_\ell L_\ell} g_{\ell,s-\tau_\ell} N_\ell(\mathrm{d}s) \right\|,$$

where $N_\ell$ is the Poisson point measure associated to $\mathcal{P}_\ell$, we have (where we use a triangle inequality for integrals):

$$\frac{K_k M_{k,\ell}}{p_k L_k}\mathbb{E}\left[\|\varepsilon_k(t)g_{k,t-\tau_k}\| \|X_\ell(t) - X_\ell(t-\tau_k)\|\right]$$

$$\leqslant \mathbb{E}\left[\int_{(t-\tau_k)^+}^{t} M_{k,\ell}\frac{\varepsilon_k(t)K_k\varepsilon_\ell(s)K_\ell}{L_k p_k p_\ell L_\ell}\|g_{k,t-\tau_k}\| \|g_{\ell,s-\tau_\ell}\| N_\ell(\mathrm{d}s)\right]$$

$$\leqslant \mathbb{E}\left[\int_{(t-\tau_k)^+}^{t} \frac{1}{2}\left(\frac{K_k^2 M_{k,\ell}}{p_k^2 L_k\sqrt{L_k L_\ell}}\|\varepsilon_k(t)g_{k,t-\tau_k}\|^2 + \frac{K_\ell^2 M_{k,\ell}}{p_\ell^2 L_\ell\sqrt{L_k L_\ell}}\|\varepsilon_\ell(s)g_{\ell,s-\tau_\ell}\|^2\right) N_\ell(\mathrm{d}s)\right].$$

For the first term, since both $\varepsilon_k(t)$ and $N_\ell(\mathrm{d}s)$ for $s$ in the integral are independent from $X(t-\tau_k)$ (and thus from $g_{k,t-\tau_k}$), and where we write $N_\ell(u,v)$ the number of clock tickings of $\mathcal{P}_\ell$ in the interval $[u,v)$, we obtain:

$$\mathbb{E}\left[\int_{(t-\tau_k)^+}^{t} \frac{1}{2}\frac{K_k^2 M_{k,\ell}}{p_k^2 L_k\sqrt{L_k L_\ell}}\|\varepsilon_k(t)g_{k,t-\tau_k}\|^2\right]$$

$$= \frac{\mathbb{E}\left[N_\ell(t-\tau_k,t)\varepsilon_k(t)\right]}{2}\frac{K_k^2 M_{k,\ell}}{p_k^2 L_k\sqrt{L_k L_\ell}}\mathbb{E}\left[\|g_{k,t-\tau_k}\|^2\right].$$

Furthermore, using our negative correlation assumption, $\mathbb{E}\left[N_\ell(t-\tau_k,t)\varepsilon_k(t)\right] \leqslant \mathbb{E}\left[N_\ell(t-\tau_k,t)\right]\mathbb{E}\left[\varepsilon_k(t)\right] = p_\ell\tau_k\mathbb{E}\left[\varepsilon_k(t)\right]$, and since $\varepsilon_k(t)$ and $g_{k,t-\tau_k}$ are independent, $\mathbb{E}\left[\varepsilon_k(t)\right]\mathbb{E}\left[\|g_{k,t-\tau_k}\|^2\right] = \mathbb{E}\left[\varepsilon_k(t)\|g_{k,t-\tau_k}\|^2\right]$.

For the second term, since the process $(\varepsilon_\ell(s)g_{\ell,s-\tau_\ell})_s$ is predictable (in the sense that it is independent from $N_u(\mathrm{d}s)$ for all $u$), we have

$$\mathbb{E}\left[\int_{(t-\tau_k)^+}^t \frac{K_\ell^2 M_{k,\ell}}{2p_\ell^2 L_\ell\sqrt{L_k L_\ell}}\|\varepsilon_\ell(s)g_{\ell,s-\tau_\ell}\|^2 N_\ell(\mathrm{d}s)\right]$$

$$= \int_{(t-\tau_k)^+}^t \frac{K_\ell^2 M_{k,\ell}}{2p_\ell^2 L_\ell\sqrt{L_k L_\ell}}\mathbb{E}\left[\|\varepsilon_\ell(s)g_{\ell,s-\tau_\ell}\|^2\right]\mathbb{E}\left[N_\ell(\mathrm{d}s)\right]$$

$$= \int_{(t-\tau_k)^+}^t \frac{K_\ell^2 M_{k,\ell}}{2p_\ell^2 L_\ell\sqrt{L_k L_\ell}}\mathbb{E}\left[\|\varepsilon_\ell(s)g_{\ell,s-\tau_\ell}\|^2\right]p_\ell\mathrm{d}s\,.$$

Hence,

$$\frac{K_k M_{k,\ell}}{p_k L_k}\mathbb{E}\left[\|\varepsilon_k(t)g_{k,t-\tau_k}\|\|X_\ell(t)-X_\ell(t-\tau_k)\|\right]$$

$$\leqslant \frac{p_\ell\tau_k K_k^2 M_{k,\ell}}{2p_k^2 L_k\sqrt{L_k L_\ell}}\mathbb{E}\left[\|\varepsilon_k(t)g_{k,t-\tau_k}\|^2\right]$$

$$+ \int_{(t-\tau_k)^+}^t \frac{K_\ell^2 M_{k,\ell}}{2p_\ell^2 L_\ell\sqrt{L_k L_\ell}}\mathbb{E}\left[\|\varepsilon_\ell(s)g_{\ell,s-\tau_\ell}\|^2\right]p_\ell\mathrm{d}s\,.$$

Combining all our elements and taking $\mathrm{d}t \to 0$, we hence have:

$$\frac{\mathrm{d}\mathbb{E}\left[G(X(t))\right]}{\mathrm{d}t} \leqslant -\sum_{k=1}^m \frac{K_k}{L_k}\left(1-\frac{K_k}{2p_k}\right)\mathbb{E}\left[\|\varepsilon_k(t)g_{k,t-\tau_k}\|^2\right]$$

$$+ \sum_{k=1}^m\sum_{\ell\sim k} \frac{p_\ell\tau_k K_k^2 M_{k,\ell}}{2p_k L_k\sqrt{L_k L_\ell}}\mathbb{E}\left[\|\varepsilon_k(t)g_{k,t-\tau_k}\|^2\right] \quad (27)$$

$$+ \sum_{k=1}^m\sum_{\ell\sim k}\int_{(t-\tau_k)^+}^t \frac{p_k K_\ell^2 M_{k,\ell}}{2p_\ell L_\ell\sqrt{L_k L_\ell}}\mathbb{E}\left[\|\varepsilon_\ell(s)g_{\ell,s-\tau_\ell}\|^2\right]\mathrm{d}s\,.$$

**Step 2** Now, introduce the following Lyapunov function:

$$\mathcal{L}_T^\gamma = \int_0^T e^{\gamma t}\mathbb{E}\left[G(X(t))-G(x^\star)\right]\mathrm{d}t\,,$$

that we wish to upper-bound by some constant, where $\gamma$ is as in (25). We have:

$$\frac{\mathrm{d}\mathcal{L}_T^\gamma}{\mathrm{d}T} = G(X(0))-G(x^\star)+\gamma\mathcal{L}_T^\gamma + \int_0^T e^{\gamma t}\frac{\mathrm{d}\mathbb{E}\left[G(X(t))\right]}{\mathrm{d}t}\mathrm{d}t\,.$$

Integrating the bound (27) on $\frac{\mathrm{d}\mathbb{E}[G(X(t))]}{\mathrm{d}t}$, we obtain, using $\int_0^T\int_{(t-\tau)^+}^t h(u)\mathrm{d}u\mathrm{d}t \leqslant \tau\int_0^T h(t)\mathrm{d}t$ for non-negative $h$:

$$\frac{\mathrm{d}\mathcal{L}_T^\gamma}{\mathrm{d}T} \leqslant G(X(0))-G(x^\star)+\gamma\mathcal{L}_T^\gamma$$

$$- \sum_{k=1}^m \frac{K_k}{L_k}\left(1-\frac{K_k}{2p_k}\right)\int_0^T e^{\gamma t}\mathbb{E}\left[\|\varepsilon_k(t)g_{k,t-\tau_k}\|^2\right]\mathrm{d}t$$

$$+ \sum_{k=1}^m A_k\int_0^T e^{\gamma t}\mathbb{E}\left[\|\varepsilon_k(t)g_{k,t-\tau_k}\|^2\right]\mathrm{d}t\,,$$

where

$$A_k = \frac{K_k^2}{2p_k L_k}\sum_{\ell\sim k}\frac{p_\ell\tau_k M_{k,\ell}}{\sqrt{L_k L_\ell}} + e^{\gamma\tau_\ell}\frac{p_\ell\tau_\ell M_{\ell,k}}{\sqrt{L_k L_\ell}}\,.$$

Remark now that we have

$$\frac{K_k^2}{2p_k L_k} + A_k \leqslant \frac{K_k}{2L_k}, \quad k \in [m].\tag{28}$$

Indeed, (28) is equivalent to

$$K_k \leqslant \frac{p_k}{1 + \sum_{\ell \sim k}\left(\frac{p_\ell \tau_k M_{k,\ell}}{\sqrt{L_k L_\ell}} + e^{\gamma \tau_\ell}\frac{p_\ell \tau_\ell M_{\ell,k}}{\sqrt{L_k L_\ell}}\right)},$$

which follows from the assumed bounds (24) on $K_k$ and the fact that $\gamma \leqslant 1/\tau_{\max}$, assumed in (25). we then have, using (28) and the fact that, by strong convexity, $G(X(t)) - G(x^\star) \leqslant \frac{1}{2\sigma}\|\nabla G(X(t))\|^2 = \frac{1}{2\sigma}\sum_{k=1}^m \|\nabla g_{k,t}\|^2$:

$$\begin{aligned}
\frac{\mathrm{d}\mathcal{L}_T^\gamma}{\mathrm{d}T} &\leqslant G(X(0)) - G(x^\star) + \gamma \mathcal{L}_T^\gamma \\
&\quad - \sum_{k=1}^m \frac{K_k}{2L_k}\int_0^{T-\tau_k} e^{\gamma(t+\tau_k)}\mathbb{E}\left[\|\varepsilon_k(t+\tau_k)g_{k,t}\|^2\right]\mathrm{d}t \\
&\leqslant G(X(0)) - G(x^\star) + \gamma \mathcal{L}_T^\gamma \\
&\quad - \min_{k\in[m]}\left(\frac{K_k \varepsilon_k e^{\gamma \tau_k}}{2L_k}\right)\int_0^{T-\tau_{\max}} e^{\gamma t}\mathbb{E}\left[\sum_{k=1}^m \|g_{k,t}\|^2\right]\mathrm{d}t \\
&\leqslant G(X(0)) - G(x^\star) + \gamma\left(\mathcal{L}_T^\gamma - \mathcal{L}_{T-\tau_{\max}}^\gamma\right),
\end{aligned}$$

where we used the assumption (25) that $\gamma \leqslant \sigma \min_{k\in[m]}\left(\frac{K_k \varepsilon_k}{L_k}\right)$.

**Step 3**  The proof is then concluded by using the following lemma, to control solutions of this delayed ordinary differential inequality.

**Lemma 1.** *Let $h : \mathbb{R} \to \mathbb{R}^+$ a differentiable function such that:*

$$\begin{aligned}
&\forall t \leqslant 0\,, h(t) = 0\,, \\
&\forall t \geqslant 0\,, h'(t) \leqslant a + b(h(t) - h(t-\tau))\,,
\end{aligned}$$

*for some positive constants $a, b, \tau$ verifying $\tau b < 1$. Then:*

$$\forall t \in \mathbb{R}\,, \quad h(t) \leqslant \frac{a(t+\tau)}{1-\tau b}\,.$$

*Proof.* Let $\delta(t) = h(t) - h(t-\tau)$. For any $t \geqslant 0$, we have:

$$\begin{aligned}
\delta(t) &= \int_{t-\tau}^t h'(s)\mathrm{d}s \\
&\leqslant \int_{t-\tau}^t (a + b\delta(s))\mathrm{d}s \\
&\leqslant \tau(a + b\sup_{s\leqslant t}\delta(s)).
\end{aligned}$$

Let $c = \frac{\tau a}{1-\tau b}$ (solution of $x = \tau(a + bx)$) and $t_0 = \inf\{t > 0 | \delta(t) \geqslant c\} \in \mathbb{R} \cup \{\infty\}$. Assume that $t_0$ is finite. Then, by continuity, $\delta(t_0) = c$ and:

$$c \leqslant \tau(a + b\sup_{s\leqslant t_0}\delta(s)) < \tau(a + bc) < c,$$

as for all $s < t_0$, $\delta(s) < c$. This is absurd, and thus $t_0$ is not finite: $\forall t > 0, \delta(t) < c$, giving us $h(t) \leqslant c(t+\tau)/\tau$ for all $t \geqslant 0$. $\square$

To conclude the proof of Theorem (4), we apply Lemma 1 to $h(T) = \mathcal{L}_T^\gamma$ with $a = G(X(0)) - G(x^\star)$, $b = \gamma$ and $\tau = \tau_{\max}$ to obtain that for all $T > 0$,

$$\mathcal{L}_T^\gamma \leqslant \left(G(X(0)) - G(x^\star)\right)\frac{T + \tau_{\max}}{1 - \tau_{\max}\gamma}\,.$$

The result of Theorem 4 follows by dividing this inequality by $\int_0^T e^{\gamma t} \mathrm{d}t = \frac{e^{\gamma T} - 1}{\gamma}$:

$$\frac{\int_0^T e^{\gamma t} \mathbb{E}\left[G(X(t)) - G(x^\star)\right] \mathrm{d}t}{\int_0^T e^{\gamma t}\left(G(X(0)) - G(x^\star)\right)\mathrm{d}t} \leqslant \frac{\gamma}{e^{\gamma T} - 1} \frac{T + \tau_{\max}}{1 - \tau_{\max}\gamma}$$

$$= \frac{\gamma T}{e^{\gamma T} - 1} \frac{1 + \tau_{\max}/T}{1 - \tau_{\max}\gamma}$$

$$\leqslant e^{-\gamma T/2} \frac{1 + \tau_{\max}/T}{1 - \tau_{\max}\gamma},$$

where we used that for $x \geqslant 0$, $\frac{e^x - 1}{x} \geqslant e^{x/2}$. $\qquad\square$

## C  Proof of Proposition 1

First, we explain how the updates of randomized gossip can also be derived from coordinate gradient descent steps. Let $A \in \mathbb{R}^{V \times E}$ be such that for all $(ij) \in E$, $Ae_{ij} = \mu_{ij}(e_i - e_j)$ for arbitrary $\mu_{ij} \in \mathbb{R}$, where $(e_{ij})_{(ij)\in E}$ and $(e_i)_{i\in V}$ are the canonical bases of $\mathbb{R}^E$ and $\mathbb{R}^V$. Then, let $g(\lambda) = \frac{1}{2}\|A\lambda\|^2$ for $\lambda \in \mathbb{R}^{E \times d}$, so that the coordinate gradient $\nabla_{ij}g(\lambda)$ writes $\nabla_{ij}g(\lambda) = \mu_{ij}((A\lambda)_i - (A\lambda)_j)$. Thus, provided that for some $\lambda_{T_k -} \in \mathbb{R}^{E \times d}$, $x_{T_k -} - \bar{x} = A\lambda_{T_k -}$, the local averaging defined in Equation (4) is equivalent to $x_{T_k} - \bar{x} = A\lambda_{T_k}$, where $\lambda_{T_k} = \lambda_{T_k -} - \frac{K_{i_k j_k}}{p_{i_k j_k}\mu_{i_k j_k}^2}\nabla_{i_k j_k}g(\lambda_{T_k -})$ for $K_{i_k j_k} = \frac{p_{i_k j_k}}{2}$. Hence, the gossip algorithm of [5] can be viewed as a simple block-coordinate gradient descent on variables $\lambda \in \mathbb{R}^{E \times d}$ indexed by the edges of the graph instead of the nodes.

*Proof of Proposition 1.* For $A \in \mathbb{R}^{V \times E}$ as defined in Section 3.1 for non-null weights $\mu_{ij}$, define the following delayed ODE:

$$\frac{\mathrm{d}\lambda_t}{\mathrm{d}t} = -\sum_{(ij)\in E} \frac{K_{ij}}{\mu_{ij}^2} e_{ij}^\top A^\top A\lambda_{t - \tau_{ij}}. \tag{29}$$

For $(y_t)$ solution of (11), if there exists $\lambda_0$ such that $A\lambda_0 = y_0$, then $y_t = A\lambda_t$ for all $t$, where $\lambda_t$ is solution of (29) initialized at the value $\lambda_0$. Then, since $AA^\top$ is the Laplacian of graph $G$ with weights $\mu_{ij}^2 > 0$, $A$ is of rank $n - 1$. For all $\lambda$, $A\lambda$ is in the orthogonal of $\mathbb{R}\mathbb{1}$ ($\mathbb{1} \in \mathbb{R}^V$ is the vector with all entries equal to 1), so that $\mathrm{Im}(A)$ is exactly the orthogonal of $\mathbb{R}\mathbb{1}$. Finally, since for $(y_t)$ a solution of (11), $y_t - (\mathbb{1}^\top y_0)\mathbb{1}$ is also solution of (11) and takes values in the orthogonal of $\mathbb{R}\mathbb{1}$, it is sufficient to prove stability of (29).

To that end, we use Theorem 1 of [26]. For $z \in \mathbb{R}^E$, let $D(z) \in \mathbb{R}^{E \times E}$ be the diagonal matrix with diagonal equal to $z$. Let $M = D(\frac{K}{\mu^2})A^\top A$. Then, the delayed ODE (29) writes as:

$$\frac{\mathrm{d}\lambda_t(ij)}{\mathrm{d}t} = -\sum_{(kl)\in E} M_{(ij),(kl)}\lambda_{t - \tau_{ij}}(kl) \quad , (ij) \in E,$$

and ODE that takes the same form as Equation (7) in [26], for $D_{(ij)}^\leftarrow = \tau_{ij}$, $D_{(ij)}^\rightarrow = 0$ and $D_{(ij)} = \tau_{ij}$, $R = E$ and with our matrix $M$. In order to ensure that $M$ is symmetric and positive semi-definite, we take $\mu_{ij}^2 = K_{ij}$, to have $M = A^\top A$. The assumptions of Theorem 1 of [26] are verified, so that the delayed ODE (29) is table if $\rho(D(\tau)M) < 1$. We then write $\rho(D(\tau)M) = \rho(D(\sqrt{\tau})A^\top AD(\sqrt{\tau}) = \rho(AD(\sqrt{\tau})(AD(\sqrt{\tau}))^\top)$, and notice that $AD(\sqrt{\tau})(AD(\sqrt{\tau}))^\top$ is the Laplacian of graph $G$ with weights $\mu_{ij}^2\tau_{ij} = K_{ij}\tau_{ij}$, concluding the proof. $\qquad\square$

## D  Proof of Theorem 1

In the proof, we use the assumed bounds $\tau_{ij}$ on actual delays in our algorithm to ensure that communications between $i$ and $j$ started at a time $t - \tau_{ij}$ induce communication updates at time $t$. Our algorithms thus behave exactly as if individual communication delays coincide with these upper bounds $\tau_{ij}$, which allows us to analyze algorithms with constant, albeit heterogeneous delays.

In contrast an analysis in discrete time would use a global iteration counter, and discrete-time delays would not be constant,making the analysis either much more involved or unable to capture the asynchronous speedup described above.

*Proof.* Theorem 1 is obtained by applying a general result on delayed coordinate descent in the continuized framework that we detail in Section B.

Specifically, we consider the function:

$$g(\lambda) = \frac{1}{2}\|A\lambda\|^2 \quad \lambda \in \mathbb{R}^{E \times d}$$

for some $A \in \mathbb{R}^{V \times E}$ such that $Ae_{ij} = \mu_{ij}(e_i - e_j)$ for all $(ij) \in E$, where we let $\mu_{ij} = -\mu_{ji}$ by convention. As in Section 3.4, there exists $\lambda \in \mathbb{R}^{E \times d}$ such that $x_0 - \bar{x} = A\lambda$. Let $(\lambda_t)_{t \geqslant 0}$ be defined with $\lambda_0 = \lambda$, and the delayed coordinate gradient steps at the clock tickings of the *P.p.p.*'s:

$$\lambda_{T_k} \leftarrow \lambda_{T_k-} - \frac{K_{i_k j_k}}{p_{i_k j_k}} \nabla_{i_k j_k} g(\lambda_{T_k - \tau_{i_k j_k}}) .$$

For all $t \geqslant 0$, we then have $x_t = \bar{x} + A\lambda_t$, where we recall that the process $(x_t)$ follows the delayed randomized gossip updates (8) of Algorithm 1. Then, for all $t \geqslant 0$, we have $g(\lambda_t) = \frac{1}{2}\|A\lambda_t\|^2 = \frac{1}{2}\|x_t - \bar{x}\|^2$.

The result of Theorem 1 follows from a control of $\mathbb{E}[g(\lambda_t)]$ that is a direct consequence of Theorem 4 in next section with the specific choices $m = |E|$ and coordinate blocks corresponding to edges. The assumptions of Theorem 4 are verified with $L_{ij} = 2\mu_{ij}^2$, $M_{(ij),(kl)} = \sqrt{L_{ij}L_{kl}}$, and strong convexity parameter $\lambda_2(\Delta_G(\nu_{ij} = \mu_{ij}^2))$ for the specific choice $\mu_{ij}^2 = K_{ij}$, as is shown in Lemmas **??, ??, ??** in the Appendix, giving us exactly Theorem 1. $\square$

**Corollary 1.** *Under the same assumptions as Theorem 1, for $(x_t)_{t \geqslant 0}$ generated with delayed randomized gossip, define $(\tilde{x}_t)_{t \geqslant 0}$ as the exponentially weighted averaging along the trajectory of $(x_t)$:*

$$\tilde{x}_t = \gamma \frac{\int_0^t e^{\gamma s} x_s \mathrm{d}s}{e^{\gamma t} - 1} .$$

*Then, for all $T \geqslant 0$,*

$$\mathbb{E}\left[\|\tilde{x}_T - \bar{x}\|^2\right] \leqslant e^{-\frac{\gamma T}{2}} \|x_0 - \bar{x}\|^2 \frac{1 + \frac{\tau_{\max}}{T}}{1 - \gamma \tau_{\max}} .$$

# E   Proof of Theorem 2

*Proof.* Following the augmented graph approach [16], for each "physical" node $i \in V$, we associate a "virtual" node $i^{\mathrm{comp}}$, corresponding to the computational unit of node $i$. We then consider the augmented graph $G^+ = (V^+, E^+)$, where $V^+ = V \cup V^{\mathrm{comp}}$ (for $V^{\mathrm{comp}} = \{i^{\mathrm{comp}}, i \in V\}$) and $E^+ = E \cup E^{\mathrm{comp}}$ (for $E^{\mathrm{comp}} = \{(ii^{\mathrm{comp}}), i \in V\}$).

For $i \in V$, function $f_i$ is then split (using $\sigma$-strong convexity) into a sum of two $\sigma/2$-strongly convex functions: $f_i = \phi_i + \phi_{i^{\mathrm{comp}}}$ where $\phi_{i^{\mathrm{comp}}}(x_i) = f_i(x_i) - \frac{\sigma}{4}\|x_i\|^2$ and $\phi_i(x_i) = \phi_{\mathrm{comm}}(x_i) = \frac{\sigma}{4}\|x_i\|^2$.

The optimization objective (1)

$$\min_{x_1 = \ldots = x_n} \frac{1}{n} \sum_{i=1}^n f_i(x_i), \quad x = (x_1, \ldots, x_n) \in \mathbb{R}^{V \times d}$$

can then be rewritten as

$$\min_{x \in \mathbb{R}^{V^+}} \left\{ F(x) = \sum_{i \in V} \phi_i(x_i) + \sum_{i^{\mathrm{comp}} \in V^{\mathrm{comp}}} \phi_{i^{\mathrm{comp}}}(x_{i^{\mathrm{comp}}}) \right\},$$

under the constraint $x_i = x_j$ for $(ij) \in E^+$. This constraint can then be rewritten as $A^\top x = 0$ for $A \in \mathbb{R}^{E^+ \times V^+}$ such that for all $(ij) \in E^+$, $Ae_{ij} = \mu_{ij}(e_i - e_j)$, as was done for network averaging,

considering the augmented graph instead of the original graph. Using Lagrangian duality, denoting $F_A^*(\lambda) := F^*(A\lambda)$ for $\lambda \in \mathbb{R}^{E^+ \times d}$ where $F^*$ is the Fenchel conjugate of $F$, we have:

$$\min_{x \in \mathbb{R}^{V^+ \times d}, x_i = x_j, (ij) \in E^+} F(x) = \max_{\lambda \in \mathbb{R}^{E \times d}} -F_A^*(\lambda).$$

Thus $F_A^*(\lambda)$ is to be minimized over the dual variable $\lambda \in \mathbb{R}^{E^+ \times d}$. The rest of the proof is divided in two steps: in the first, we derive the updates of the *DDO* algorithm from coordinate gradient descent steps on dual variables, and in the second step we apply Theorem 4 to prove rates of convergence for these coordinate gradient descent steps on function $F_A^*$.

The partial derivative of $F_A^*$ with respect to coordinate $(ij) \in E^+$ of $\lambda \in \mathbb{R}^{E^+ \times d}$ reads:

$$\nabla_{ij} F_A^*(\lambda) = \mu_{ij}(\nabla \phi_i^*((A\lambda)_i) - \nabla \phi_j^*((A\lambda)_j)).$$

Consider then the following step of coordinate gradient descent for $F_A^*$ on coordinate $(i_k j_k) \in E^+$ of $\lambda$, performed when edge $(i_k j_k)$ is activated at iteration $k$ (corresponding to time $t_k$):

$$\lambda_{t_k} = \lambda_{t_k-} - \frac{1}{(2\sigma^{-1})\mu_{i_k j_k}^2} \nabla_{i_k j_k} F_A^*(\lambda_{t_k - \tau_{i_k j_k}}), \tag{30}$$

corresponding to an instantiation of delayed coordinate gradient descent in the continuized framework, on function $F_A^*$, for *P.p.p.* of intensities $(p_{ij})$ for $(ij) \in E$ and $p_i^{\mathrm{comp}}$ for $(ii^{\mathrm{comp}}) \in E^{\mathrm{comp}}$. Denoting $v_t = A\lambda_t \in \mathbb{R}^{V^+ \times d}$ for $t \geqslant 0$, we obtain the following formula for updating coordinates $i_k, j_k$ of $v$ when $i_k j_k$ activated, *irrespectively of the choice of $\mu_{ij}$ in matrix $A$*:

$$
\begin{aligned}
v_{t_k, i_k} &= v_{t_k-, i_k} - \frac{\nabla \phi_{i_k}^*(v_{t_k - \tau_{i_k j_k}, i_k}) - \nabla \phi_{j_k}^*(v_{t_k - \tau_{i_k j_k}, j_k})}{2\sigma^{-1}}, \\
v_{t_k, j_k} &= v_{t_k-, j_k} + \frac{\nabla \phi_{i_k}^*(v_{t_k - \tau_{i_k j_k}, i_k}) - \nabla \phi_{j_k}^*(v_{t_k - \tau_{i_k j_k}, j_k})}{2\sigma^{-1}}.
\end{aligned}
\tag{31}
$$

Such updates can be performed locally at nodes $i$ and $j$ after communication between the two nodes (if $(ij)$ is a 'physical edge'), or locally (if $(ij)$ is 'virtual edge'). We refer in the sequel to this scheme as the Coordinate Descent Method. While $\lambda \in \mathbb{R}^{E \times d}$ is a dual variable defined on the edges, $v \in \mathbb{R}^{n \times d}$ is also a dual variable, but defined on the nodes. The *primal surrogate* of $v$ is defined as $x = \nabla F^*(v)$ *i.e.* $x_i = \nabla f_i^*(v_i)$ at node $i$. It can hence be computed with local updates on $v$. The decentralized updates of Algorithm 3 (computational updates in Algorithm 2, communication updates in Algorithm 1) are then direct consequences of Equation (31).

The last step of the proof consists in applying Theorem 4 in order to obtain Theorem 2. The function $F_A^*$ we introduced satisfies the assumptions of Theorem 4 with coordinate blocks corresponding to edges $E^+$: The regularity assumptions are satisfied with smoothness parameter $L_{ij} = 8\mu_{ij}^2 \sigma^{-1}$ and local Lipschitz coefficients $M_{(ij),(kl)} = \sqrt{L_{ij} L_{kl}}$ for any $(ij), (kl) \in E^+$, as shown in Lemmas **??** and **??** in the Appendix. $F_A^*$ is moreover $\sigma$-strongly convex[6] with $\sigma$ derived using Lemmas **??** and **??**, and the weights associated to matrix $A$ are chosen so that $\mu_{ij}^2 = \frac{\varepsilon_{ij} K_{ij} \sigma}{2\mu_{ij}^2}$.

Finally, the output of the algorithm at node $i$ is the primal surrogate of variable $x_i(t)$ (associated to $\phi_i$), which is equal to $\nabla \phi_i(x_i(t)) = \frac{\sigma}{2} x_i(t)$.

$\square$

# F   Proof of Theorem 3

*Proof.* The algorithm under capacity constraints is obtained by applying coordinate gradient descent in the continuized framework to the same dual problem as in Section 4, but with random variables "$\varepsilon_k(t)$" that are not taken constant equal to 1. Here, for $(ij) \in E$ and $t \in \mathcal{P}_{ij}$, we have

$$\varepsilon_{ij}(t) = \mathbb{1}_{\left\{n_{ij}(t) < q_{ij}, \, n_i^{\mathrm{comm}}(t) < q_i^{\mathrm{comm}}, \, n_j^{\mathrm{comm}}(t) < q_j^{\mathrm{comm}}\right\}},$$

---

[6]In fact, it is strongly convex on the orthogonal of $\mathrm{Ker}A$, which suffices for us to conclude since the dynamics are restricted to this subspace.

while for $i \in V$ and $t \in \mathcal{P}_i^{\text{comp}}$,

$$\varepsilon_{ii^{\text{comp}}}(t) = 1_{\left\{n_i^{\text{comp}} < q_i^{\text{comp}}\right\}} .$$

We apply Theorem 4 as in the proof of Theorem 2, leading to the same stability conditions on the step sizes $K_{ij}, K_i^{\text{comp}}$, while the rate of convergence is multiplied by a lower bound $\varepsilon$ on all $\mathbb{E}\left[\varepsilon_{ij}(t)\right]$ and $\mathbb{E}\left[\varepsilon_{ii^{\text{comp}}}(t)\right]$. Let us finally compute such a lower bound $\varepsilon$.

For $(ij) \in E$, $n_{ij}(t)$ is stochastically dominated by $Z_{ij}$ a Poisson random variable of parameter $p_{ij}\tau_{ij}$, while $n_i^{\text{comm}}(t)$ and $n_j^{\text{comm}}(t)$ are respectively dominated by $Z_i$ and $Z_j$, Poisson random variables of parameters $\tau_{ij}\sum_{k\sim i} p_{ki}$ and $\tau_{ij}\sum_{\ell\sim j} p_{\ell j}$, so that:

$$\mathbb{E}\left[\varepsilon_{ij}(t)\right] \geqslant \mathbb{P}\left(Z_{ij} < q_{ij}, Z_i < q_i^{\text{comm}}, Z_j < q_j^{\text{comm}}\right)$$
$$\geqslant 1 - \mathbb{P}(Z_{ij} \geqslant q_{ij}) - \mathbb{P}(Z_i \geqslant q_i^{\text{comm}}) - P(Z_j \geqslant q_j^{\text{comm}}) .$$

We now prove that $\mathbb{P}(Z_{ij} \geqslant q_{ij}), \mathbb{P}(Z_i \geqslant q_i^{\text{comm}}), P(Z_j \geqslant q_j^{\text{comm}})$ are all inferior to $1/6$. For $\mathbb{P}(Z_{ij} \geqslant q_{ij})$, using that for $Z$ a Poisson variable of parameter $\mu$ and $x \geqslant 0$,

$$\mathbb{P}(Z_\mu \geqslant \mu + x) \leqslant e^{\frac{-x^2}{\mu+x}} ,$$

we have $\mathbb{P}(Z_{ij} \geqslant q_{ij}) \leqslant e^{-\frac{(q_{ij}-p_{ij}\tau_{ij})^2}{q_{ij}}} \leqslant e^{-2(1-1/c)^2}$ if $q_{ij} \geqslant 2$, and this quantity is equal to $1/6$, by definition of $c$. Then, if $q_{ij} = 1$, using $\mathbb{P}(Z_\mu \geqslant 1) = 1 - e^{-\mu}$, we have that $\mathbb{P}(Z_{ij} \geqslant q_{ij}) \leqslant p_{ij}\tau_{ij} \leqslant 1/c \leqslant 1/6$. We proceed in the same way for $\mathbb{P}(Z_i \geqslant q_i^{\text{comm}}), P(Z_j \geqslant q_j^{\text{comm}})$. Hence, $\mathbb{E}\left[\varepsilon_{ij}(t)\right] \geqslant 1/2$ under our assumptions on the Poisson intensities. Similarly, we prove that $\mathbb{E}\left[\varepsilon_{ii^{\text{comp}}}(t)\right] \geqslant 1/2$, and this concludes the proof. $\qquad\square$

