# OpenReview forum: "Asynchronous speedup in decentralized optimization"
_NeurIPS.cc/2022/Workshop/Federated_Learning — FL-NeurIPS 2022 Poster_

### Official Review · Reviewer_8PP7 · 2022-10-17
**Good paper. Theorem 1 discussion doesn't not comment on the condition number**

This paper is about minimizing the sum of convex in a decentralized manner. In these settings, nodes may fail to carry out updates due to system/network failures. The paper proposes an algorithm allowing nodes to perform decentralized updates asynchronously. The problem is relevant, especially for large networks where devices and network conditions are heterogeneous.

Strong points:
- The paper is well written, and the contribution is clearly stated in Theorem 1.
- The problem is relevant to federated learning, although the problem and solution apply to more general settings.

Weak points:
- In Eq. (3), the authors ask: “how robust to stragglers can decentralized algorithms be?”. I’m not sure how well Theorem 1 answers that question.
- The convergence bounds in Theorem 1 do not depend on the condition number $\kappa$. Is it the condition number contained in another variable? I recommend adding a discussion after Theorem 1.
- Why do the authors express optimality in terms of primal variable difference instead of $f(x_k) - f^\star$? Is this a common practice in decentralized/consensus problems?

---

### Official Review · Reviewer_72kG · 2022-10-17
**Borderline, leaning towards accept**

This paper studies a randomized gossip model accounting for heterogeneity in communication and processing times.

Using the term "asynchronous" to refer to the model and algorithm studied in this paper may be confusing and misleading. Although there is a notion of staleness and delay, these delays are somehow synchronized, in that when nodes i and j communicate, the send their messages at the same times, and they each update their models at exactly the same time. This seems to be very disconnected from reality, where typically one would want to use asynchrony precisely to decouple event times at different nodes (so i doesn't need to wait for j if j is slow). It also seems like it would be challenging, or require additional overhead, to actually ensure that the execution of an distributed algorithm adheres to the description in Algorithm 1 (since nodes i and j effectively need to synchronize to be sure they implement the update at the same time. Moreover, assuming that delays follow a Possion point process is convenient for analysis, but difficult to validate or justify in practice. (For reference, what is more typically reflective of asynchronous algorithms is the case where delays are simply assumed to be bounded, rather than following any particular distribution; the delays may be different at different nodes, e.g., as captured in the well-studied framework studied in [Tsitsiklis, Bertsekas, and Athans (1986)](http://www.mit.edu/~jnt/Papers/J014-86-asyn-grad.pdf).)

Apart from these more practical concerns, the theory introduced in the paper is interesting and elegant, and extensions thereof may ultimately be more broadly relevant.

Finally, the connection to federated learning (the main subject of the workshop) is indirect. Although such decentralized algorithms may be useful for cross-silo FL, there remains much work to be done to provide the sorts of privacy and security guarantees that are expected in most FL applications (privacy and security being *the* main reason to use FL in the first place...).

Overall, for these reasons, I believe the paper is borderline, but I lean towards accepting it since I expect some of the FL-NeurIPS community may be interested in the results, and the analysis technique may also be of broader interest.

---

### Official Review · Reviewer_ShUm · 2022-10-18
**The authors present a new way to analyze asynchronous algorithms in decentralized networks with delays using a continued framework. The approach yields a characterization of convergence time and its dependency on heterogeneous delays in the network.**

The authors present a new way to analyze asynchronous algorithms in decentralized networks with delays using a continued framework. The approach yields a characterization of convergence time and its dependency on heterogeneous delays in the network.

It is not clear to me whether Assumption 1 is feasible in practice. Additionally, this assumption has a vague impact on the improvement of analysis.

---

### Decision · Program_Chairs · 2022-10-20

Accept (Poster)